# Discovery of 42 genome-wide significant loci associated with dyslexia

Catherine Doust[1], Pierre Fontanillas [2], Else Eising [3], Scott D. Gordon [4], Zhengjun Wang [5], Gökberk Alagöz[3], Barbara Molz [3], 23andMe Research Team*, Quantitative Trait Working Group of the GenLang Consortium*, Beate St Pourcain [3,6,7], Clyde Francks [3,6], Riccardo E. Marioni [8], Jingjing Zhao[5], Silvia Paracchini[9], Joel B. Talcott [10], Anthony P. Monaco [11], John F. Stein[12], Jeffrey R. Gruen [13], Richard K. Olson [14,15], Erik G. Willcutt[14,15], John C. DeFries[14,15], Bruce F. Pennington[16], Shelley D. Smith[17], Margaret J. Wright [18], Nicholas G. Martin [4], Adam Auton, Timothy C. Bates [1], Simon E. Fisher [3,6] and Michelle Luciano [1]✉

Reading and writing are crucial life skills but roughly one in ten children are affected by dyslexia, which can persist into adulthood. Family studies of dyslexia suggest heritability up to 70%, yet few convincing genetic markers have been found. Here we performed a genome-wide association study of 51,800 adults self-reporting a dyslexia diagnosis and 1,087,070 controls and identified 42 independent genome-wide significant loci: 15 in genes linked to cognitive ability/educational attainment, and 27 new and potentially more specific to dyslexia. We validated 23 loci (13 new) in independent cohorts of Chinese and European ancestry. Genetic etiology of dyslexia was similar between sexes, and genetic covariance with many traits was found, including ambidexterity, but not neuroanatomical measures of language-related circuitry. Dyslexia polygenic scores explained up to 6% of variance in reading traits, and might in future contribute to earlier identification and remediation of dyslexia.

The ability to read is crucial for success at school and access to employment, information and health and social services, and is related to attained socioeconomic status[1]. Dyslexia is a neurodevelopmental disorder characterized by severe reading difficulties, present in 5–17.5% of the population, depending on diagnostic criteria[2,3]. It often involves impaired phonological processing (the decoding of sound units, or phonemes, within words) and frequently co-occurs with psychiatric and other developmental disorders[4], especially attention-deficit hyperactivity disorder

[1]Department of Psychology, University of Edinburgh, Edinburgh, UK. [2]23andMe, Inc., Sunnyvale, CA, USA. [3]Language and Genetics Department, Max Planck Institute for Psycholinguistics, Nijmegen, the Netherlands. [4]Genetic Epidemiology Laboratory, QIMR Berghofer Medical Research Institute, Brisbane, Queensland, Australia. [5]School of Psychology, Shaanxi Normal University and Shaanxi Key Research Center of Child Mental and Behavioral Health, Xi'an, China. [6]Donders Institute for Brain, Cognition and Behaviour, Radboud University, Nijmegen, the Netherlands. [7]MRC Integrative Epidemiology Unit, University of Bristol, Bristol, UK. [8]Centre for Genomic and Experimental Medicine, Institute of Genetics and Cancer, University of Edinburgh, Edinburgh, UK. [9]School of Medicine, University of St Andrews, St Andrews, UK. [10]Institute of Health and Neurodevelopment, Aston University, Birmingham, UK. [11]Office of the President, Tufts University, Medford, MA, USA. [12]Department of Physiology, Anatomy and Genetics, Oxford University, Oxford, UK. [13]Departments of Pediatrics and Genetics, Yale Medical School, New Haven, CT, USA. [14]Department of Psychology and Neuroscience, University of Colorado, Boulder, CO, USA. [15]Institute for Behavioral Genetics, University of Colorado, Boulder, CO, USA. [16]Department of Psychology, University of Denver, Denver, CO, USA. [17]Department of Neurological Sciences, College of Medicine, University of Nebraska Medical Center, Omaha, NE, USA. [18]Queensland Brain Institute, University of Queensland, Brisbane, Queensland, Australia. *Lists of authors and their affiliations appear at the end of the paper. ✉e-mail: michelle.luciano@ed.ac.uk

(ADHD)[5,6] and speech and language disorders[7,8]. Dyslexia may represent the low extreme of a continuum of reading ability, a complex multifactorial trait with heritability estimates ranging from 40% to 80%[9,10]. Identifying genetic risk factors not only aids increased understanding of the biological mechanisms, but may also expand diagnostic capabilities, facilitating earlier identification of individuals prone to dyslexia and co-occurring disorders for specific support.

Previous genome-wide investigations of dyslexia have been limited to linkage analyses of affected families[11] or modest ($n < 2,300$ cases) association studies of diagnosed children and adolescents[12]. Candidate genes from linkage studies show inconsistent replication, and genome-wide association studies (GWAS) have not found significant associations, although *LOC388780* and *VEPH1* were supported in gene-based tests[12]. Larger cohorts are vital for increasing sensitivity to detect new genetic associations of small effect. Here, we present the largest dyslexia GWAS to date, with 51,800 adults self-reporting a dyslexia diagnosis and 1,087,070 controls, all of whom are research participants with the personal genetics company 23andMe, Inc. We validate our association discoveries in independent cohorts, provide functional annotations of significant variants (mainly single-nucleotide polymorphisms (SNPs)) and potential causal genes, and estimates of SNP-based heritability. Lastly, we investigate genetic correlations with reading and related skills, health, socioeconomic, and psychiatric measures, and evaluate the evidence for previously implicated dyslexia candidate genes in our well-powered results.

## Results

### Genome-wide associations

The full dataset included 51,800 (21,513 males, 30,287 females) participants responding 'yes' to the question 'Have you been diagnosed with dyslexia?' (cases) and 1,087,070 (446,054 males, 641,016 females) participants responding 'no' (controls). Participants were aged 18 years or over (mean ages of cases and controls were 49.6 years (s.d. 16.2) and 51.7 years (s.d. 16.6), respectively). We identified 42 independent genome-wide significant associated loci ($P < 5 \times 10^{-8}$) and 64 loci with suggestive significance ($P < 1 \times 10^{-6}$) (Fig. 1 and Supplementary Table 1). Genomic inflation was moderate ($\lambda_{GC} = 1.18$) and consistent with polygenicity (see Q–Q plot, Extended Data Fig. 1). We also performed sex-specific GWAS and age-specific GWAS (younger or older than 55 years) because dyslexia prevalence was higher in our younger (5.34% in 20- to 30-year-olds) than older (3.23% in 80- to 90-year-olds) participants. These subsample analyses showed high consistency with the main GWAS (of the full sample). Genetic correlation estimated by linkage disequilibrium (LD) score regression (LDSC) was 0.91 (95% confidence intervals (CI): 0.86–0.96; $P = 8.26 \times 10^{-253}$) in males and females, and 0.97 (95% CI: 0.91–1.02; $P = 2.32 \times 10^{-268}$) between younger and older adults.

Of the 17 genome-wide significant variants in the female GWAS (Extended Data Fig. 2), all but four (rs61190714, rs4387605, rs12031924 and rs57892111) were significant in the main GWAS and, of these four, three were in LD with an SNP that approached significance ($P < 3.3 \times 10^{-7}$ or smaller) in the main analysis. Intergenic SNP rs57892111 (located between *TFAP2B* and *PKHD1* on chromosome 6p) was not among the significant or suggestive SNPs of the main analysis, and so may represent a female-specific variant. There is no evidence from existing GWAS that this SNP is associated with any other human trait. Of the six genome-wide significant variants in the male GWAS (Extended Data Fig. 3), all were significant in the main GWAS.

In the main GWAS, all significant variants were autosomal, except rs5904158 at Xq27.3 (for regional association plots, see Supplementary Fig. 1). A total of 17 index variants were in high LD with published (genome-wide significant) associated SNPs in the NHGRI GWAS Catalog[13] (15 were associated with cognitive/educational traits; Supplementary Tables 1 and 2). Thus, a total of 27 associated loci showed no evidence of published genome-wide associations with traits expected to overlap with dyslexia (for example, educational attainment, cognitive ability) and were considered new (Table 1).

Of 38 associated loci (the 4 remaining were tagged by indels unavailable in validation cohorts), 3 (rs13082684, rs34349354 and rs11393101) were significant at a Bonferroni-corrected level ($P < 0.05/38$) in the GenLang consortium GWAS meta-analysis of reading ($n = 33,959$) and spelling ($n = 18,514$) ability[14]. At $P < 0.05$, 18 were associated in GenLang, 3 in the NeuroDys case-control GWAS[12] ($n = 2,274$ cases), and 5 in the Chinese Reading Study (CRS) of reading accuracy and fluency ($n = 2,270$; Supplementary Note) (Table 1 and Supplementary Tables 3–6).

Gene-based tests identified 173 significantly associated genes (Supplementary Table 7) but no significantly enriched biological pathways (Supplementary Table 8). We estimated the LDSC liability-scale SNP-based heritability of dyslexia to be $h^2_{SNP} = 0.152$ (standard error = 0.006) using the 23andMe sample prevalence of 5%, and $h^2_{SNP} = 0.189$ (standard error = 0.008) using a 10% prevalence of dyslexia, which is more typical of the general population[2,3].

### Fine-mapping and functional annotations

Within the credible variant set (Supplementary Table 1), missense variants were the most common (55%) of the coding variants; Extended Data Figure 4 summarizes all predicted variant effects. Predicted deleterious variants by SIFT (Sorting Intolerant From Tolerant) score were identified in *R3HCC1L*, *SH2B3*, *CCDC171*, *C1orf87*, *LOXL4*, *DLAT*, *ALG9* and *SORT1*. Within the credible variant set, no genes were especially intolerant to functional variation (smallest LoFtool (Loss-of-Function) percentile was 0.39). For the 42 associated loci, the most probable gene targets of each were estimated by the Overall V2G (Variant-to-Gene) score from OpenTargets (Supplementary Table 9). Two index variants (missense variant rs12737449 (*C1orf87*) and rs3735260 (*AUTS2*)) could be causal because they had combined annotation dependent depletion (CADD) scores suggestive of deleteriousness to gene function according to Kircher et al.[15] (Supplementary Table 10). The *AUTS2* variant RegulomeDB rank of 2b indicated a regulatory role; its chromatin state supported location at an active transcription start site[16,17].

Of the 173 significant genes from genome-wide gene-based tests in MAGMA (see Supplementary Table 11 for their functions), 129 could be functionally annotated (Supplementary Table 12). Protein-coding and noncoding sequences are actively conserved in approximately three-quarters of these genes, 63% are more intolerant to variation than average and 33% are intolerant to loss-of-function mutations. Gene property analysis for general tissues and 13 brain tissues confirmed the importance of the brain and specific brain regions (Supplementary Tables 13 and 14). Levels of brain expression for 125 of the 173 significant genes from gene-based tests could be mapped in FUMA and are shown in Supplementary Table 15. A total of 20 genes showed high general brain expression levels and, of these, 3 (*PPP1R1B*, *NPM1* and *WASF3*) were located near significant SNP associations. Of the 12 brain regions assessed, gene expression was generally highest in the cerebellar hemisphere, cerebellum, and cerebral cortex, consistent with the results of gene property analysis.

### Partitioned heritability

SNP-based heritability of dyslexia partitioned by functional annotation showed significant enrichment for conserved regions and H3K4me1 clusters (Supplementary Table 16 and Extended Data Fig. 5). There was enrichment in genes expressed in the frontal cortex, cortex and anterior cingulate cortex ($P < 4.17 \times 10^{-3}$) (Supplementary Table 17 and Extended Data Fig. 6), but not for brain cell type (Supplementary Table 18 and Extended Data Fig. 7). Enrichment was seen in enhancer and promoter regions, identified by the presence of H3K4me1 and H3K4me3 chromatin marks, respectively, in multiple central nervous system (CNS) tissues (Supplementary Tables 19 and 20 and Extended Data Figs. 8 and 9). Reading, an offshoot of spoken language, is a uniquely human

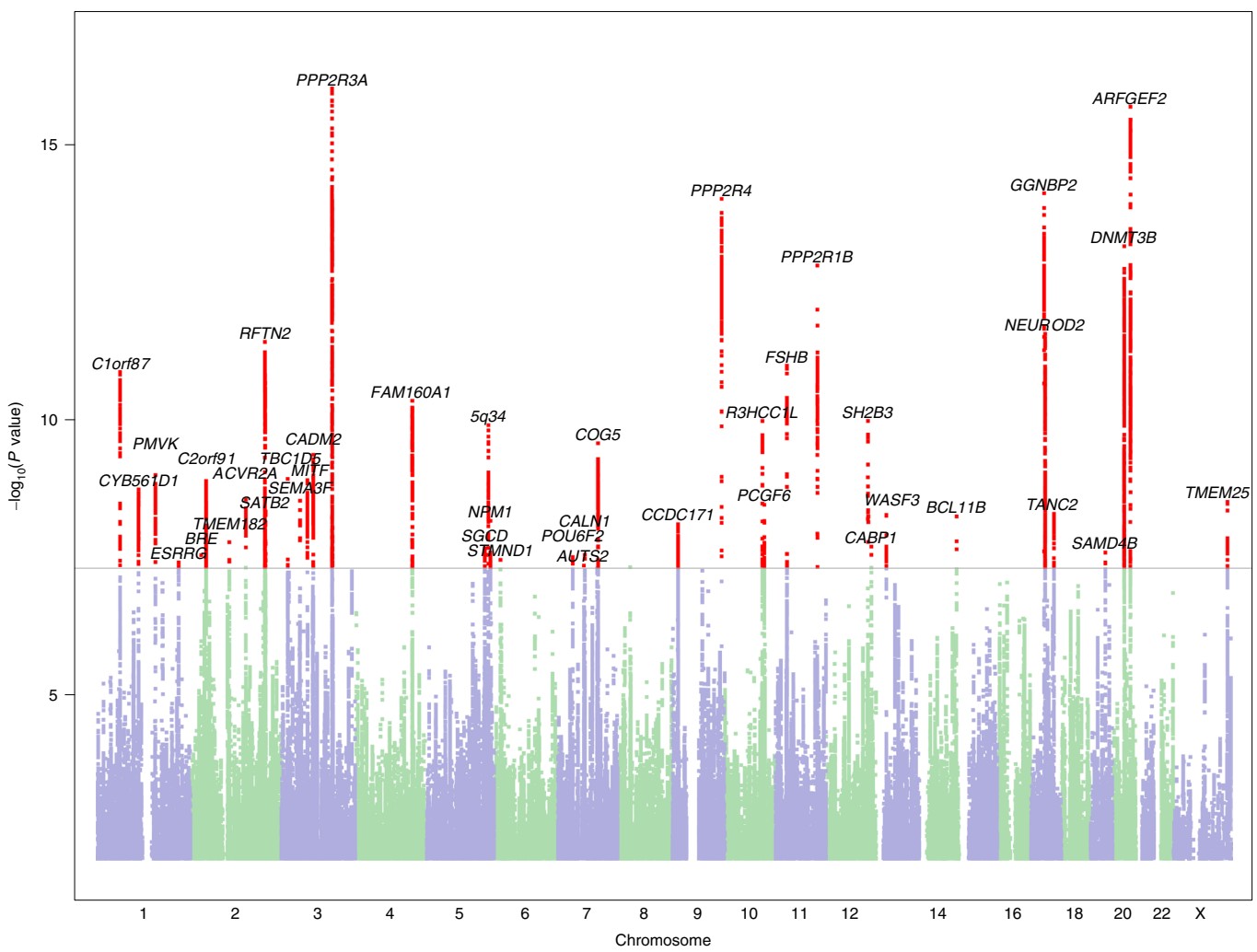

**Fig. 1 | Manhattan plot of the genome-wide association analysis of dyslexia.** The y axis represents the $-\log_{10}P$ value for association of SNPs with self-reported dyslexia diagnosis from 51,800 individuals and 1,087,070 controls. The threshold for genome-wide significance ($P < 5 \times 10^{-8}$) is represented by a horizontal grey line. Genome-wide significant variants in the 42 genome-wide significant loci are red. Variants located within a distance of <250 kb of each other are considered as one locus.

trait, but there was no enrichment for a range of annotations related to human evolution spanning the last 30 million to 50,000 years[18] (Supplementary Table 21).

## Genetic correlations and LDSC

Genetic correlations were estimated for 98 traits (Fig. 2 and Supplementary Table 22), including reading and spelling measures, from GenLang (Fig. 3), and brain subcortical structure volumes, total cortical surface area and thickness from the Enhancing Neuro Imaging Genetics through Meta-Analysis (ENIGMA) consortium. A total of 63 traits showed genetic correlations with dyslexia at the Bonferroni-corrected significance threshold ($P < 0.05/98$; Fig. 2). Genetic correlations ($r_g$) with quantitative reading and spelling measures ranged from −0.70 to −0.75 (lowest 95% CI of −0.60, highest 95% CI of −0.86), and were −0.62 (95% CI: −0.50, −0.74) and −0.45 (95% CI: −0.26, −0.64) with phoneme awareness and nonword repetition measures, respectively. The childhood/adolescent performance (nonverbal) intelligence quotient (IQ) $r_g$ was lower (−0.19; 95% CI: −0.08, −0.30) than that for adult verbal-numerical reasoning[19] (−0.50; 95% CI: −0.45, −0.55) but similar to that for childhood IQ[20] (−0.32; 95% CIs: −0.21, −0.43) and educational attainment[21] (−0.22; 95% CI: −0.15, −0.29). Traits showing positive $r_g$ included jobs involving heavy manual work[21] (0.40; (95% CI: 0.34, 0.45)), work-related/vocational qualifications[21] (0.50; 95% CI:

0.41, 0.59), ADHD[22] (0.53; 95% CI: 0.29, 0.77), equal use of right and left hands[21] (0.38; 95% CI: 0.19, 0.57) and pain measures[21] (average = 0.31; 95% CI: 0.21, 0.41). Of the 11 ENIGMA measures tested, only intracranial volume was significantly correlated with dyslexia ($r_g$ = −0.14; 95% CI: −0.06, −0.22). Targeted investigation of 80 structural neuroimaging measures from UK Biobank, including surface-based morphometry and diffusion-weighted imaging for brain circuitry linked to language, were nonsignificant at a Bonferroni-corrected significance level for number of independent traits. Phenotype independence was estimated by spectral decomposition of the phenotypic correlation matrix implied by the bivariate LDSC intercept from GWAS summary statistics of these traits, using the PhenoSpD toolkit[23] (Supplementary Table 23).

## Polygenic score analyses

Dyslexia polygenic scores (PGS) based on the 23andMe dyslexia GWAS were computed in four independent cohorts and, overall, higher PGS were associated with lower reading and spelling accuracy (Supplementary Table 24). In two Australian population-based samples (1,647 adolescents, 1,163 adults), the dyslexia PGS explained up to 3.6% of variance in the reading and spelling measures, being most predictive of lower performance on tests of nonword reading, an index of phonological decoding. Dyslexia PGS did not correlate with scores on tests of nonword repetition (considered a marker of phonological short-term

**Table 1 | New SNP associations with dyslexia, including gene-based results, eQTL status, expression in brain and validation in three independent cohorts (GenLang Consortium, CRS and NeuroDys)**

| Cytoband | SNP | Effect allele | Frequency | Odds Ratio | GWAS P | Gene(s) | Most probable gene | Validation cohort (P uncorrected for multiple testing) |
|---|---|---|---|---|---|---|---|---|
| chr1q21.3 | rs4845687 | A | 0.56 | 1.044 | $1.1 \times 10^{-9}$ | KCNN3, PMVK | PMVK[ab] | GenLang (0.02) |
| chr2q22.3 | rs497418 | A | 0.38 | 1.043 | $3.0 \times 10^{-9}$ | ACVR2A | AC062032.1[c] | GenLang (0.009) |
| chr2q33.1 | rs72916919 | G | 0.51 | 1.049 | $4.1 \times 10^{-12}$ | **RFTN2** | MARS2[a] | NeuroDys (0.02), GenLang (0.02) |
| chr3p12.1 | rs10511073 | A | 0.37 | 1.046 | $4.6 \times 10^{-10}$ | **CADM2** | **CADM2**[a] | GenLang (0.02) |
| chr3q22.3 | rs13082684 | A | 0.24 | 1.069 | $1.0 \times 10^{-16}$ | **PPP2R3A** | **PPP2R3A** (intron)[a] | GenLang (0.0004); not in CRS |
| chr6p22.3 | rs2876430 | T | 0.34 | 1.041 | $3.7 \times 10^{-8}$ | ATXN1, STMND1 | STMND1 | GenLang (0.04) |
| chr7p14.1 | rs62453457 | G | 0.48 | 1.039 | $3.3 \times 10^{-8}$ | **POU6F2** | **POU6F2** | CRS (0.04) |
| chr7q11.22 | rs3735260 | G | 0.08 | 1.075 | $4.7 \times 10^{-8}$ | **AUTS2** | **AUTS2** | GenLang (0.02) |
| chr7q11.22 | rs77059784 | G | 0.97 | 1.123 | $3.0 \times 10^{-8}$ | **CALN1** | **CALN1** | GenLang (0.02); not in CRS |
| chr9q34.11 | rs9696811 | C | 0.69 | 1.069 | $1.1 \times 10^{-16}$ | **PPP2R3A** | AL158151.4[abc] | GenLang (0.03) |
| chr11q23.1 | rs138127836 | A | 0.65 | 1.056 | $1.7 \times 10^{-13}$ | **PPP2R1B** | **PPP2R1B** (intron)[ab] | GenLang (0.02) |
| chr17q23.3 | rs72841395[c] | C | 0.77 | 1.049 | $5.4 \times 10^{-9}$ | **TANC2** | **TANC2**[a] | GenLang (0.005) |
| chrXq27.3 | rs5904158 | GTA | 0.65 | 1.037 | $3.3 \times 10^{-8}$ | TMEM257, CXorf51B[b] | AL109653.3[c] | GenLang (0.02); not in NeuroDys/CRS |
| chr2q12.1 | rs367982014 | CAAT | 0.29 | 1.045 | $1.8 \times 10^{-8}$ | **TMEM182** | **MFSD9**[a] | Not available |
| chr3p24.3 | rs373178590 | G | 0.51 | 1.046 | $1.3 \times 10^{-9}$ | **TBC1D5** | **TBC1D5** (intron)[a] | Not available |
| chr10q24.33 | rs34732054 | C | 0.57 | 1.045 | $3.7 \times 10^{-9}$ | **PCGF6** | **USMG5**[a] | Not available |
| chr13q12.13 | rs375018025 | CA | 0.57 | 1.044 | $5.6 \times 10^{-9}$ | CDK8, **WASF3** | **WASF3** | Not available |
| chr1p32.1 | rs12737449 | G | 0.85 | 1.070 | $1.4 \times 10^{-11}$ | **C1orf87** | **C1orf87** (missense)[a] | Not significant |
| chr2p23.2 | rs1969131 | T | 0.17 | 1.053 | $3.0 \times 10^{-8}$ | **BABAM2** | **BABAM2** | Not significant |
| chr3q26.33 | rs7625418 | C | 0.21 | 1.056 | $4.3 \times 10^{-9}$ | PEX5L, TTC14 | TTC14[a] | Not significant |
| chr3p13 | rs13097431 | G | 0.58 | 1.044 | $1.3 \times 10^{-9}$ | **MITF** | **MITF**[a] | Not significant |
| chr5q33.3 | rs867009 | G | 0.36 | 1.041 | $2.3 \times 10^{-9}$ | **SGCD** | **SGCD**[a] | Not significant |
| chr9p22.3 | rs3122702 | T | 0.5 | 1.041 | $8.3 \times 10^{-9}$ | **CCDC171** | **CCDC171**[ab] | Not significant |
| chr10q24.2 | rs10786387 | C | 0.68 | 1.049 | $1.1 \times 10^{-10}$ | **CRTAC1, R3HCC1L** | **R3HCC1L**[a] | Not significant |
| chr11p14.1 | rs676217 | G | 0.37 | 1.050 | $1.1 \times 10^{-11}$ | KCNA4, **FSHB** | ARL14EP[ab] | Not significant |
| chr19q13.2 | rs60963584 | A | 0.89 | 1.065 | $2.7 \times 10^{-8}$ | **GMFG, SAMD4B** | **SAMD4B**[a] | Not significant |
| chr20q11.21 | rs4911257 | C | 0.39 | 1.055 | $7.5 \times 10^{-14}$ | **DNMT3B** | **DNMT3B** (intron)[ab] | Not significant |

Statistics for each variant are from the 23andMe GWAS (see Supplementary Table 1 for all 42 significant variants). Genes that are significant in gene-based tests are set in bold. Multi-allelic effect alleles represent insertions. The most probable gene is that most likely to be causal based on genetic and functional genomic data tied to the tag SNP (https://platform.opentargets.org/). [a]eQTL. [b]eQTL linked to brain expression. [c]Not available in gene-based results.

memory). In developmental cohorts enriched for reading difficulties, the dyslexia PGS explained 3.7% (UKdys; $n = 930$) and 5.6% (CLDRC; $n = 717$) of variance in word recognition tests.

**Analyses of dyslexia associations from the literature**
Of 75 previously reported dyslexia associations, none showed genome-wide significance in our analyses (Supplementary Table 25). Of these targeted variants, 19 (in ATP2C2, CMIP, CNTNAP2, DCDC2, DIP2A, DYX1C1, FOXP2, KIAA0319L and PCNT) showed association surviving Bonferroni correction that accounted for LD ($P < 0.05/68.7$). In gene-based tests of 14 candidate genes from the literature[24,25],

association at a Bonferroni level ($P < 0.05/14$) was seen for KIAA0319L ($P = 1.84 \times 10^{-4}$) and ROBO1 ($P = 1.53 \times 10^{-3}$) (Supplementary Table 26). The CNTNAP2 association approached corrected replication-level significance ($P = 0.004$). Targeted gene set analysis of three pathways previously implicated in dyslexia (Supplementary Table 27) showed replication-level support ($P = 2.00 \times 10^{-3}$) for the axon guidance pathway (comprising 216 genes).

## Discussion
In the largest GWAS of dyslexia to date (>50,000 self-reported diagnoses), we identified 42 significant independent loci. Of these,

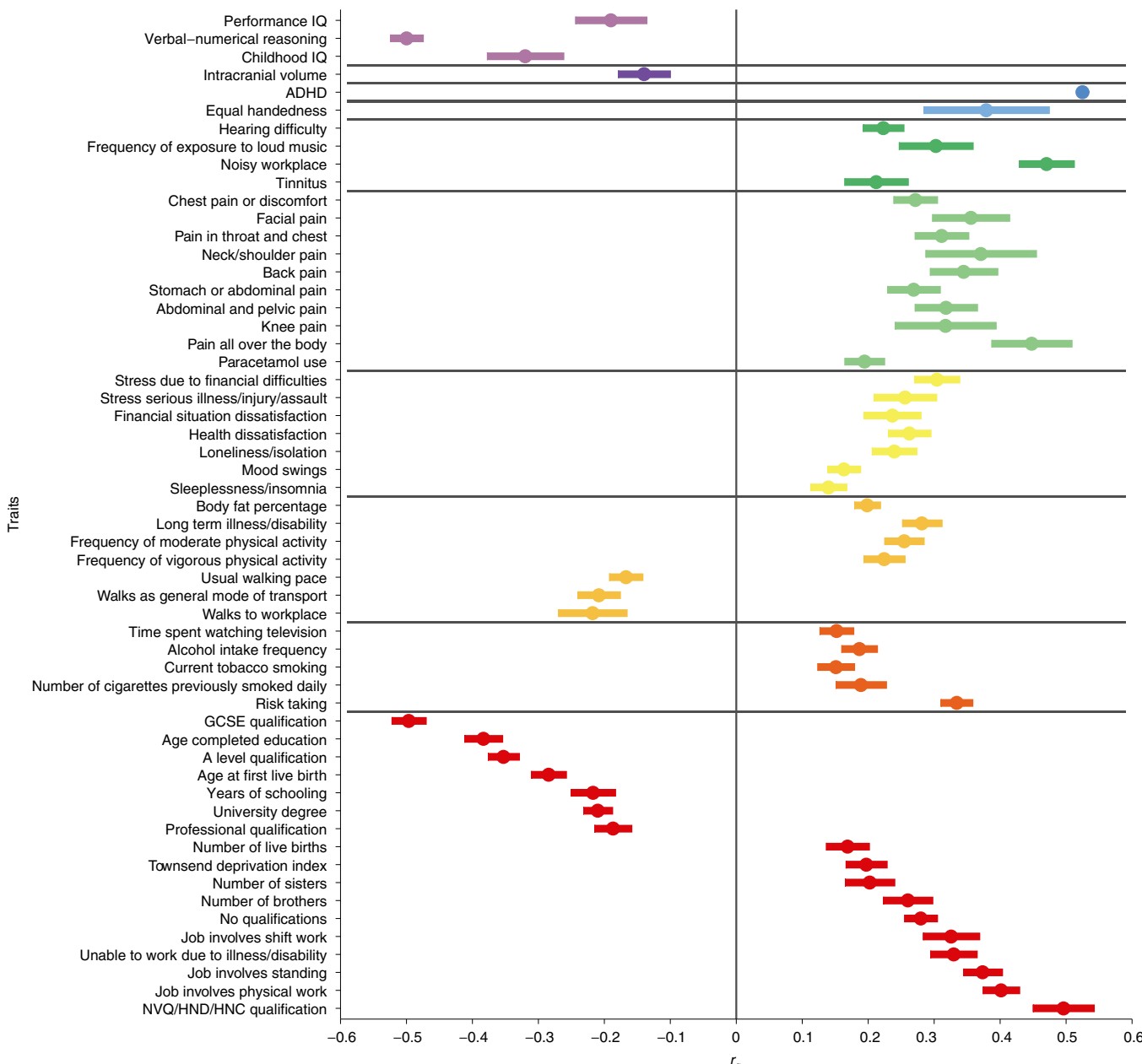

**Fig. 2 | Genetic correlations of dyslexia with other phenotypes.** Significant ($P < 5 \times 10^{-4}$) genetic correlations ($r_g$) between self-reported dyslexia diagnosis from 23andMe and other phenotypes from the LD Hub database and Enhancing Neuro Imaging Genetics Through Meta-Analysis (ENIGMA). We tested 98 traits but present only those that were significant after Bonferroni correction. Center points represent genetic correlations, and error bars represent standard errors around the estimate; exact values can be found in Supplementary Table 22. The vertical line indicates a genetic correlation of zero, and the horizontal lines divide groups of related traits. GCSE, General Certificate of Secondary Education; HNC, Higher National Certificate; HND, Higher National Diploma; NVQ, National Vocational Qualification.

27 represent new associations that have not been uncovered in GWAS of related cognitive traits; 12 of the new associations were validated in the GenLang consortium GWAS meta-analysis of reading/spelling in English and other European languages[14], and 1 in a Chinese language cohort. Of the significant SNPs, 36% overlapped with variants from general cognitive ability GWAS, consistent with twin studies that find that genetic variation in reading disability is explained by general and reading-specific cognitive ability[10]. Similar to other complex traits, and consistent with high polygenicity, each significant locus showed small effects (odds ratios (ORs) ranging from 1.04 to 1.12). Our estimated SNP-based heritability of 19% (assuming a 10% dyslexia population prevalence) was equal to that reported in a smaller GWAS[12], but lower than heritability estimates from twin studies (40–80%)[26,27]. This

difference may be due partly to effects of rare and structural variants[28], which have been implicated in reading and related traits[29,30].

Whereas *AUTS2* has been implicated in autism[31], intellectual disability[32] and dyslexia[33], the variant we uncovered (rs3735260) represents the strongest *AUTS2* SNP association with a neurodevelopmental trait to date. Amongst our findings were other known neurodevelopmental genes, such as *TANC2* (implicated in language delay and intellectual disability[34,35]) and, especially, *GGNBP2* (linked to neurodevelopmental delay[36] and autism[37]) with variant rs34349354 supported in all our validation cohorts. However, rs34349354 is also associated with cognitive performance[38], and based on expression quantitative trait loci (eQTL) evidence is more likely linked to *ZNHIT3*, colocalizing with molecular QTLs (opentargets.org). Notably, none of the more established

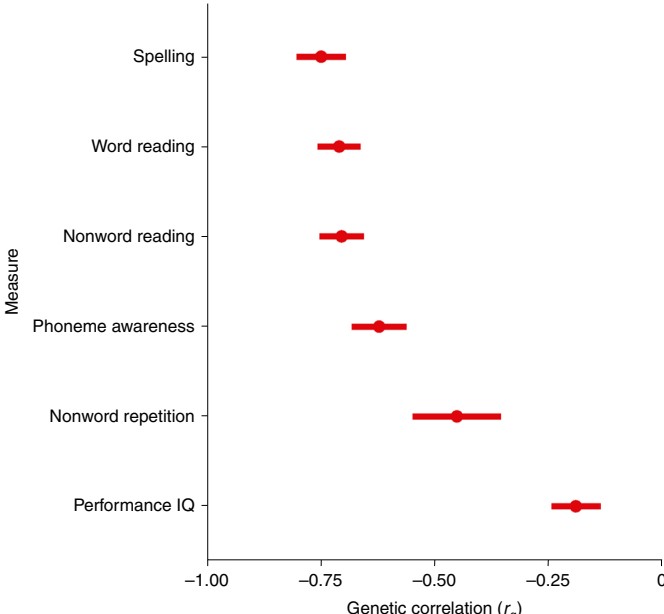

**Fig. 3 | Genetic correlations between dyslexia and measures of reading, language and nonverbal IQ.** Genetic correlations ($r_g$) between self-reported dyslexia diagnosis from 23andMe and measures of reading, language and performance (nonverbal) IQ in the GenLang consortium. Center points represent genetic correlations estimated in LDSC, and error bars represent standard errors around the estimate; exact values can be found in Supplementary Table 22.

candidate genes for dyslexia approached genome-wide significance in our results.

Like other human complex traits, partitioning of SNP-based heritability revealed enrichment in conserved regions[39]. We further observed enrichment in the histone mark H3K4me1 (which has also been reported for ASD[40]), and at H3K4me1 and H3K4me3 clusters in the CNS (marking enhancers and promoters, respectively). Since reading/writing systems are built on our capacities for spoken language, it is plausible that evolutionary changes on the human lineage helped shape the underlying genetic architecture[41]. However, we did not find enrichment of significant associations for curated annotations spanning different periods of hominin prehistory.

Our self-reported dyslexia diagnosis binary trait showed strong negative genetic correlations with quantitative reading and spelling measures, supporting the validity of this measure in the 23andMe cohort, and suggesting that reading skills and disorder are not qualitatively distinct. The positive genetic correlation between hearing difficulties and dyslexia is consistent with genetic correlations reported for childhood reading skill[42], suggesting that hearing problems at an early age could affect acquisition of phonological processing skills.

Dyslexia showed moderately negative genetic correlations with adult verbal-numerical reasoning, but there was a lack of a strong genetic correlation of dyslexia with (nonverbal) performance IQ. This would be consistent with phenotypic observations that individuals with dyslexia are disadvantaged on verbal IQ tests[43]. Educational attainment correlations were also not strong, which might reflect school adjustments and other support that counteract disadvantage in academic learning.

There was little evidence of common genetic variation in dyslexia being related to interindividual differences in subcortical volumes, or structural connectivity and morphometry for brain regions implicated in language processing in adults. Thus, the phenotypic correlations previously reported between dyslexia and aspects of neuroanatomy may in large part reflect environmental shaping of the brain, perhaps through the process of reading itself[44]. Left-handedness

and ambidexterity show small genetic overlap with each other[45] yet are both phenotypically linked to neurodevelopmental disorders/cognitive abilities[46,47]. We report a significant genetic correlation between dyslexia and self-reported equal hand use, but not left-handedness, supporting theories linking ambidexterity and dyslexia[48].

Dyslexia and ADHD[5,6] often co-occur (24% reporting ADHD in our cases versus 9% in controls), and we show a moderate genetic correlation between the two, potentially reflecting shared endophenotypes like deficits in working memory and attention[49]. Although we did not find significant genetic correlations between dyslexia and ASD, the GWAS for the latter encompassed diverse neurodevelopmental phenotypes, including subgroups with varying educational attainment and IQ[40]. Genetic correlations with pain-related traits suggest that individuals with dyslexia may have a lower threshold for pain perception. Links between pain and other neurodevelopmental disorders have been reported[50,51].

Dyslexia polygenic scores were correlated with lower achievement on reading and spelling tests in population-based and reading-disorder enriched samples, especially for nonword reading, a measure of phonological decoding that is typically impaired in dyslexia. Polygenic scores could become a valuable tool to help identify children with a propensity for dyslexia, enabling learning support before development of reading skills. However, a limitation of our study is the potential for collider bias arising from sample selection (that is, people without dyslexia and from higher socioeconomic positions), which we were unable to quantify; thus, care should be taken in future research when using polygenic scores based on many variants[52].

In summary, we report 42 new independent genome-wide significant loci associated with dyslexia, 27 of which have not been associated with cognitive-educational traits and should be prioritized for follow up as dyslexia candidates. Functional annotation of the variants highlights the importance of conserved and enhancer regions of the genome for this trait. Dyslexia shows positive genetic correlations with ADHD, vocational qualifications, physical occupations, ambidexterity and pain perception, and negative correlations with academic qualifications and cognitive ability; family-based methods are needed to dissociate pleiotropic and causal effects.

## Online content

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

## 23andMe Research Team

Stella Aslibekyan[2], Adam Auton[2], Elizabeth Babalola[2], Robert K. Bell[2], Jessica Bielenberg[2], Katarzyna Bryc[2], Emily Bullis[2], Daniella Coker[2], Gabriel Cuellar Partida[2], Devika Dhamija[2], Sayantan Das[2], Sarah L. Elson[2], Teresa Filshtein[2], Kipper Fletez-Brant[2], Pierre Fontanillas[2], Will Freyman[2], Pooja M. Gandhi[2], Karl Heilbron[2], Barry Hicks[2], David A. Hinds[2], Ethan M. Jewett[2], Yunxuan Jiang[2], Katelyn Kukar[2], Keng-Han Lin[2], Maya Lowe[2], Jey McCreight[2], Matthew H. McIntyre[2], Steven J. Micheletti[2], Meghan E. Moreno[2], Joanna L. Mountain[2], Priyanka Nandakumar[2], Elizabeth S. Noblin[2], Jared O'Connell[2], Aaron A. Petrakovitz[2], G. David Poznik[2], Morgan Schumacher[2], Anjali J. Shastri[2], Janie F. Shelton[2], Jingchunzi Shi[2], Suyash Shringarpure[2], Vinh Tran[2], Joyce Y. Tung[2], Xin Wang[2], Wei Wang[2], Catherine H. Weldon[2], Peter Wilton[2], Alejandro Hernandez[2], Corinna Wong[2] and Christophe Toukam Tchakouté[2]

## Quantitative Trait Working Group of the GenLang Consortium

Filippo Abbondanza[9], Andrea G. Allegrini[19], Till F. M. Andlauer[20,21], Cathy L. Barr[22,23,24], Timothy C. Bates[1], Manon Bernard[25], Kirsten Blokland[23], Milene Bonte[26], Dorret I. Boomsma[27,28,29], Thomas Bourgeron[30,31], Daniel Brandeis[32,33,34,35], Manuel Carreiras[36,37,38], Fabiola Ceroni[39,40], Valéria Csépe[41,42], Philip S. Dale[43], John C. DeFries[14,15], Peter F. de Jong[44], Jean Francois Démonet[45], Eveline L. de Zeeuw[28], Else Eising[3], Yu Feng[46], Simon E. Fisher[3,6], Marie-Christine J. Franken[47], Clyde Francks[3,6], Margot Gerritse[3], Alessandro Gialluisi[20,48], Scott D. Gordon[4], Jeffrey R. Gruen[13], Sharon L. Guger[49], Marianna E. Hayiou-Thomas[50], Juan Hernández-Cabrera[51], Jouke-Jan Hottenga[28], Charles Hulme[52], Philip R. Jansen[53,54,55], Juha Kere[56,57], Elizabeth N. Kerr[49,58,59], Tanner Koomar[60], Karin Landerl[61,62], Gabriel T. Leonard[63], Zhijie Liao[64], Maureen W. Lovett[23,59], Michelle Luciano[1], Heikki Lyytinen[65], Nicholas G. Martin[4], Angela Martinelli[9], Urs Maurer[66], Jacob J. Michaelson[60], Nazanin Mirza-Schreiber[67], Kristina Moll[68], Anthony P. Monaco[11], Angela T. Morgan[69,70,71], Bertram Müller-Myhsok[20,72], Dianne F. Newbury[40], Markus M. Nöthen[73], Richard K. Olson[14,15], Silvia Paracchini[9], Tomas Paus[74,75,76], Zdenka Pausova[25,77], Craig E. Pennell[78,79,80], Bruce F. Pennington[16], Robert J. Plomin[19], Kaitlyn M. Price[23,24,46], Franck Ramus[81], Sheena Reilly[69,82], Louis Richer[83], Kaili Rimfeld[19], Gerd Schulte-Körne[68], Chin Yang Shapland[7,84], Nuala H. Simpson[85], Shelley D. Smith[17], Margaret J. Snowling[85,86], Beate St Pourcain[3,6,7], John F. Stein[87], Lisa J. Strug[88,89], Joel B. Talcott[10], Henning Tiemeier[53,90], J. Bruce Tomblin[91], Dongnhu T. Truong[13], Elsje van Bergen[27,28,92], Marc P. van der Schroeff[93,94], Marjolein Van Donkelaar[3], Ellen Verhoef[3], Carol A. Wang[78,79], Kate E. Watkins[85], Andrew J. O. Whitehouse[95], Karen G. Wigg[46], Erik G. Willcutt[14,15], Margaret Wilkinson[23], Margaret J. Wright[18] and Gu Zhu[4]

[19]Social, Genetic and Developmental Psychiatry Centre, Institute of Psychiatry, Psychology and Neuroscience, King's College London, London, UK. [20]Translational Research in Psychiatry, Max Planck Institute of Psychiatry, Munich, Germany. [21]Department of Neurology, Klinikum rechts der Isar, School of Medicine,Technical University of Munich, Munich, Germany. [22]Division of Experimental and Translational Neuroscience, Krembil Research Institute, University Health Network, Toronto, Ontario, Canada. [23]Program in Neuroscience and Mental Health, Hospital for Sick Children, Toronto, Ontario, Canada. [24]Department of Physiology, University of Toronto, Toronto, Ontario, Canada. [25]Departments of Physiology and Nutritional Sciences, Hospital for Sick Children, Toronto, Ontario, Canada. [26]Department of Cognitive Neuroscience and Maastricht Brain Imaging Center, Faculty of Psychology and Neuroscience, Maastricht University, Maastricht, the Netherlands. [27]Netherlands Twin Register, Amsterdam, the Netherlands. [28]Department of Biological Psychology, Vrije Universiteit Amsterdam, Amsterdam, the Netherlands. [29]Amsterdam Reproduction and Development (AR&D) Research Institute, Amsterdam, the Netherlands. [30]Human Genetics and Cognitive Functions Unit, Institut Pasteur, Paris, France. [31]CNRS UMR 3571, Université de Paris, Paris, France. [32]Department of Child and Adolescent Psychiatry and Psychotherapy, Psychiatric Hospital, University of Zurich, Zurich, Switzerland. [33]Zurich Center for Integrative Human Physiology (ZIHP), University of Zurich and ETH Zurich, Zurich, Switzerland. [34]Neuroscience Center Zurich, University of Zurich and ETH Zurich, Zurich, Switzerland. [35]Department of Child and Adolescent Psychiatry and Psychotherapy, Central Institute of Mental Health, Medical Faculty Mannheim, Heidelberg University, Mannheim, Germany. [36]Basque Center on Cognition, Brain and Language (BCBL), Donostia-San Sebastian, Spain. [37]Ikerbasque, Basque Foundation for Science, Bilbao, Spain. [38]Lengua Vasca y Comunicación, University of the Basque Country (UPV/EHU),

Bilbao, Spain. [39]Department of Pharmacy and Biotechnology, University of Bologna, Bologna, Italy. [40]Faculty of Health and Life Sciences, Oxford Brookes University, Oxford, UK. [41]Brain Imaging Centre, Research Centre for Natural Sciences, Budapest, Hungary. [42]Multilingualism Doctoral School, Faculty of Modern Philology and Social Sciences, University of Pannonia, Veszprém, Hungary. [43]Department of Speech and Hearing Sciences, University of New Mexico, Albuquerque, NM, USA. [44]Department of Child Development and Education, University of Amsterdam, Amsterdam, the Netherlands. [45]Leenaards Memory Centre, Department of Clinical Neurosciences Lausanne University Hospital (CHUV), University of Lausanne, Lausanne, Switzerland. [46]Genetics and Development Division, Krembil Research Institute, University Health Network, Toronto, Ontario, Canada. [47]Department of Otorhinolaryngology, Erasmus University Medical Centre, Rotterdam, the Netherlands. [48]Department of Epidemiology and Prevention, IRCCS Istituto Neurologico Mediterraneo Neuromed, Pozzilli, Italy. [49]Department of Psychology, Hospital for Sick Children, Toronto, Ontario, Canada. [50]Department of Psychology, University of York, York, UK. [51]Departamento de Psicología Clínica Psicobiología y Metodología, Universidad de La Laguna, Santa Cruz de Tenerife, Spain. [52]Department of Education, University of Oxford, Oxford, UK. [53]Department of Child and Adolescent Psychiatry/Psychology, Erasmus University Medical Center, Rotterdam, the Netherlands. [54]Department of Complex Trait Genetics, Center for Neurogenomics and Cognitive Research, Amsterdam Neuroscience, VU University, Amsterdam, the Netherlands. [55]Department of Human Genetics, VU Medical Center, Amsterdam UMC, Amsterdam, the Netherlands. [56]Department of Biosciences and Nutrition, Karolinska Institutet, Stockholm, Sweden. [57]Stem Cells and Metabolism Research Program, University of Helsinki, and Folkhälsan Research Center, Helsinki, Finland. [58]Department of Neurology, Hospital for Sick Children, Toronto, Ontario, Canada. [59]Department of Paediatrics, The University of Toronto, Toronto, Ontario, Canada. [60]Department of Psychiatry, University of Iowa, Iowa City, IA, USA. [61]Institute of Psychology, University of Graz, Graz, Austria. [62]BioTechMed-Graz, Graz, Austria. [63]Cognitive Neuroscience Neurology and Neurosurgery, Montreal, Quebec, Canada. [64]Department of Psychology, University of Toronto, Toronto, Ontario, Canada. [65]Department of Psychology, University of Jyväskylä, Jyväskylä, Finland. [66]Department of Psychology, The Chinese University of Hong Kong, Hong Kong, China. [67]Institute of Neurogenomics, Helmholtz Zentrum München, Munich, Germany. [68]Department of Child and Adolescent Psychiatry, Psychosomatics, and Psychotherapy, LMU University Hospital Munich, Munich, Germany. [69]Speech and Language, Murdoch Children's Research Institute, Melbourne, Victoria, Australia. [70]Department of Audiology and Speech Pathology, University of Melbourne, Melbourne, Victoria, Australia. [71]Speech Pathology Department, Royal Children's Hospital, Melbourne, Victoria, Australia. [72]Department of Health Science, University of Liverpool, Liverpool, UK. [73]Institute of Human Genetics, University Hospital of Bonn, Bonn, Germany. [74]Department of Psychiatry, University of Toronto, Toronto, Ontario, Canada. [75]Departments of Psychiatry and Neuroscience and Centre Hospitalier Universitaire Sainte Justine, University of Montreal, Montreal, Quebec, Canada. [76]Department of Psychology, University of Toronto, Toronto, Ontario, Canada. [77]Hospital for Sick Children, Toronto, Ontario, Canada. [78]School of Medicine and Public Health, College of Health, Medicine and Wellbeing, University of Newcastle, Newcastle, New South Wales, Australia. [79]Mothers and Babies Research Centre, Hunter Medical Research Institute, Newcastle, New South Wales, Australia. [80]Maternity and Gynaecology, John Hunter Hospital, Newcastle, New South Wales, Australia. [81]Laboratoire de Sciences Cognitives et Psycholinguistique, Ecole Normale Supérieure, PSL University, EHESS, CNRS, Paris, France. [82]Menzies Health Institute Queensland, Griffith University, Gold Coast, Queensland, Australia. [83]Department of Health Sciences, Université du Québec à Chicoutimi, Chicoutimi, Quebec, Canada. [84]Population Health Sciences, University of Bristol, Bristol, UK. [85]Department of Experimental Psychology, University of Oxford, Oxford, UK. [86]St John's College, University of Oxford, Oxford, UK. [87]Department of Physiology, Anatomy and Genetics, Oxford University, Oxford, UK. [88]Departments of Statistical Sciences and Computer Science and Division of Biostatistics, University of Toronto, Toronto, Ontario, Canada. [89]Program in Genetics and Genome Biology and The Centre for Applied Genomics, Hospital For Sick Children, Toronto, Ontario, Canada. [90]Harvard T.H. Chan School of Public Health, Boston, MA, USA. [91]Communication Sciences and Disorders, University of Iowa, Iowa City, IA, USA. [92]Research Institute LEARN!, Vrije Universiteit Amsterdam, Amsterdam, the Netherlands. [93]Department of Otolaryngology, Head and Neck Surgery, Erasmus MC, Rotterdam, the Netherlands. [94]Generation R Study Group, Erasmus MC, Rotterdam, the Netherlands. [95]Telethon Kids Institute, The University of Western Australia, Perth, Western Australia, Australia.

## Methods

### GWAS participants

Participants were drawn from the customer base of 23andMe, Inc., a consumer genetics company. Participants provided informed consent and participated in the research online, under a protocol approved by the external AAHRPP-accredited IRB, Ethical and Independent Review Services (www.eandireview.com). They included 51,800 (21,513 male, 30,287 female) participants who responded 'yes' to the question 'Have you been diagnosed with dyslexia?' (cases) and 1,087,070 (446,054 male, 641,016 female) participants who responded 'no' (controls). Age ranged from 18 to 110 years, with the prevalence of dyslexia higher for younger participants (5.34% in those aged 20–30 years) than older participants (3.23% in those aged 80–90 years). The negative linear relationship between dyslexia prevalence and participant age was expected given that screening for specific learning difficulties has only become commonplace in more recent decades. Moreover, this aligns with findings from the subsample (4.3%) of participants who reported age of diagnosis: younger participants were diagnosed at an earlier age (for example, 9.7 years (±4.7) for 20- to 30-year-olds) than older participants (for example, 22.4 years (±17.8) for 80- to 90-year-olds). The prevalence of dyslexia in our sample was similar for women (4.51%) and men (4.6%), although the slightly higher prevalence in males in this very large sample was statistically significant ($P < 8.7 \times 10^{-6}$). Such a prevalence lies at the lower end of the range typically reported in the US population[3] and might represent the more severe cases of dyslexia given that a formal diagnosis was required; additionally, people with dyslexia might opt out of survey research that requires reading, further restricting the sample range.

### Genotyping and imputation

DNA was extracted from saliva samples and genotyped on one of five genotyping platforms by the National Genetics Institute (NGI). In the present analysis, only participants with European ancestry were included. Details about the genotyping arrays, quality control of samples and ancestry derivation can be found in Fontanillas et al.[53] and the Supplementary Note. Phased genotypes were imputed to a combined reference panel of the 1000 Genomes Phase 3 haplotypes (May 2015) and the UK10K imputation reference panel using Minimac3 (see Das et al.[54]).

### Association analysis

Association analysis was performed on genotyped and imputed SNP dosage data using logistic regression and assuming an additive model of allelic effects. For X-chromosome analysis, male genotypes were treated as homozygous diploid. Covariates included age, age squared, gender, the first five ancestry principal components and genotype platform. SNP significance was evaluated by a likelihood ratio test, and genome-wide significance was determined as $P < 5 \times 10^{-8}$ (suggestive significance level as $P < 1 \times 10^{-6}$). Only reliably imputed SNPs ($r^2 > 0.80$) and those with minor allele frequency (MAF) > 0.01 are presented ($n = 7,995,923$). We define associated regions by first identifying all variants with $P < 5 \times 10^{-8}$, then grouping these variants into regions separated by gaps of at least 250 kb. Index variants are the variants with smallest $P$ value within each associated region. We use the same approach for regions with suggestive associations, but by first identifying all variants with $P < 10^{-5}$. Subsidiary genome-wide association analysis of separate male ($n = 21,513$ cases, 446,054 controls) and female ($n = 30,287$ cases, 641,016 controls) groups, and younger (below 55 years; $n = 30,763$ cases, 582,276 controls) and older (55 and above; $n = 21,037$ cases, 504,794 controls) groups was performed. The latter was to check whether reliability of diagnosis (assumed to be higher in the younger sample whose recall of diagnosis should be better and who would have been exposed to greater levels of dyslexia screening) affected the GWAS signal.

We also looked to independently validate our genome-wide significant variants within (1) a published GWAS meta-analysis of 2,274 dyslexia cases from nine European countries representing six different languages (NeuroDys) by Gialluisi et al.[55]; (2) a population sample (Chinese Reading Study; CRS) of children measured on quantitative traits of reading accuracy and reading fluency ($n = 2,270$; described in the Supplementary Note), and; (3) within the GenLang quantitative trait GWAS meta-analysis of word reading (up to $n = 33,959$) and spelling (up to $n = 18,514$) skills measured in cohorts of children and adolescents from Europe, the United States and Australia, and representing seven European languages, of which English was the most common[14].

### Genomic control

Top SNPs are reported from the more conservative GWAS results adjusted for genomic control (Fig. 1, Extended Data Figs. 1–4, and Supplementary Tables 1, 2, 9 and 10), whereas downstream analyses (including gene-set analysis, enrichment and heritability partitioning, genetic correlations, polygenic prediction, candidate gene replication) are based on GWAS results without genomic control.

### Gene-based analyses

The GWAS results were used to calculate gene-based $P$ values for association with dyslexia by performing the gene analysis in MAGMA v.1.08 (ref. [56]) through the FUMA interface[57] using standard settings. In total, 19,039 genes were tested, and $P$ values were judged based on a Bonferroni-corrected significance threshold of $P < 2.63 \times 10^{-6}$. We also performed gene set analyses for association of biological pathways (all available gene ontology (GO) terms and curated gene sets from the Molecular Signatures Database (MsigDB)[58,59]) with dyslexia in MAGMA through the FUMA interface. The total number of pathways tested was 15,486, and $P$ values were judged based on a Bonferroni-corrected significance threshold of $P < 3.23 \times 10^{-6}$.

### Biological annotations

Genome-wide significant variants and nearby gene(s) were annotated using external reference data and evaluated for functional or regulatory impact. A 99% credible set of potentially causal variants for SNPs in significant regions was based on approximate Bayes factor (ABFs)[60] assuming a prior variance of 0.1, and using the method of Maller et al.[61] to define these sets. Variant effect prediction of these was done in ENSEMBL (release 104)[62]. For genome-wide significant variants, we considered: gene context (whether a variant is intergenic or located within a specific functional region within a gene locus); deleteriousness (Combined Annotation Dependent Depletion (CADD) score); functionality (RegulomeDB (RDB) category); chromatin state (minimum and common 15-core chromatin state); and SNP-trait associations reported in the NHGRI GWAS Catalog[13].

For each variant, the most probable gene target was identified using the Open Target Genetics portal[63], which draws on evidence from QTL and chromatin interaction experiments, functional predictions and distance from a gene's transcription start site. For genome-wide significant genes, we considered: loss-of-function intolerance (probability of loss-of-function Intolerance (pLI) score); variation intolerance (residual variation intolerance score, RVIS); variation intolerance in noncoding regions (noncoding RVIS, ncRVIS); evolutionary constraint of noncoding regions (noncoding genomic evolutionary rate profiling (ncGERP) score); evolutionary constraint of protein-coding regions (protein-coding genomic evolutionary rate profiling (pcGERP) score); deleteriousness across noncoding regions (noncoding CADD (ncCADD) score); combined functionality of variants in noncoding regions (noncoding genome-wide annotation of variants (ncGWAVA) score); and expression in 12 brain tissues (amygdala, anterior cingulate cortex, caudate basal ganglia, cerebellar hemisphere, cerebellum, cortex, frontal cortex, hippocampus, hypothalamus, nucleus accumbens basal ganglia, putamen basal ganglia and substantia nigra). All annotations were obtained through FUMA[57] except RVIS, ncGERP, pcGERP, ncCADD and ncGWAVA, which were taken from

Petrovski et al.[64]. Details of each annotation including original sources are in the Supplementary Note.

## Partitioned heritability

We partitioned SNP heritability of dyslexia using stratified LDSC, as described by Finucane et al.[39], to determine whether SNPs that share the greatest proportion of the heritability are also clustered in specific functional categories in the genome. Overall, we performed 266 different tests, which would give a very conservative Bonferroni-corrected significance level of $1.88 \times 10^{-4}$, but because there will be overlap among annotation groups, we also report corrections to significance within different classes of annotation, each of which we now describe. Partitioning was performed for the 24 main functional annotations defined by Finucane et al.[39]. LD scores, regression weights and allele frequencies are from European ancestry samples and were retrieved from https://alkesgroup.broadinstitute.org/LDSCORE. Heritability estimates were considered statistically significant if the $P$ value surpassed an $\alpha$ level of $2.08 \times 10^{-3}$, derived by Bonferroni correction based on 24 tests.

We also estimated the enrichment for heritability of dyslexia for tissue-specific annotations, while controlling for the annotations in the baseline model, including gene expression in three brain cell types, gene expression in 12 brain regions, and chromatin marks H3K4me1 and H3K4me3 in multiple tissues (108 and 114, respectively) since these marks are enriched at enhancers[65] and promoters[66], respectively. Enrichment is the proportion of SNP heritability divided by the proportion of SNPs. For the brain cell types, we estimated enrichment for heritability of dyslexia for genes expressed in neurons, astrocytes, and oligodendrocytes using data from Cahoy et al.[67]. Enrichments were considered statistically significant if the $P$ value surpassed an $\alpha$ level of 0.017, derived by Bonferroni correction based on three tests. The gene expression data used to estimate the enrichment of heritability in genes expressed in certain brain regions was from the GTEx database[68], and the Bonferroni-derived $\alpha$ level for enrichment was $4.17 \times 10^{-3}$ (based on 12 tests). Chromatin annotations include data from the Roadmap Epigenomics consortium[17] and EN-TEx[69,70]. For H3K4me1, the Bonferroni-derived $\alpha$ level for enrichment was $4.63 \times 10^{-4}$ (based on 108 tests) and, for H3K4me3, the Bonferroni-derived $\alpha$ level for enrichment was $4.39 \times 10^{-4}$ (based on 114 tests).

**Evolutionary annotations.** Although reading and writing is a human cultural invention, it builds on fundamental pathways involved in language processing. Therefore, we investigated whether annotations related to human evolution were significantly enriched for heritability of dyslexia by applying an evolutionary analysis pipeline adapted from Tilot et al.[18]. These analyses capture a range of periods in an evolutionary timeframe on the lineage that led to humans, from approximately 30 million years ago to 50,000 years ago.

Enrichment of heritability was estimated in adult brain human gained enhancers (HGEs)[71], fetal brain HGEs[72], ancient selective sweep regions[73], Neanderthal-introgressed SNPs[74] and Neanderthal-depleted regions[75] (see Supplementary Note for a description of each annotation); and controlled for using the baselineLD v.2 model from Gazal et al.[76]. Heritability enrichment in human adult and fetal HGEs were additionally controlled for adult and fetal brain active regulatory elements from the Roadmap Epigenomics resource[17]. Active regulatory elements were defined using chromHMM[16]. Enrichment $P$ values were judged by an $\alpha$ level of $10^{-2}$, derived by Bonferroni correction based on five tests.

## Genetic correlations

### Genetic correlations within the 23andMe GWAS of dyslexia. Genetic correlation between self-reported dyslexia diagnosis in males and females, and between younger (<55 years old) and older (≥55 years old) adults was calculated using LDSC[77,78].

**Genetic correlations of dyslexia with other traits.** We present the pairwise genetic correlation of dyslexia with 98 traits. Summary statistics for most of these traits are publicly available through LD Hub[77–79]—a centralized database and web interface that automates the LDSC regression analysis pipeline. A selection of brain magnetic resonance imaging measures obtained from the ENIGMA-3 consortium[80–83], and measures of reading and spelling accuracy, and performance IQ from the GenLang Consortium[14] were analyzed locally using LDSC. Word reading accuracy in GenLang was measured by the number of correct words read aloud from a list in a time restricted or unrestricted fashion. Examples of tools that include this measure are Test of Word Reading Efficiency (TOWRE), the British Ability Scales (BAS) and the Wide Range Achievement Test (WRAT). Spelling accuracy in GenLang was measured by the number of words correctly spelled orally or in writing. The words were dictated as single words or in a sentence. Examples of tools that include this measure are the BAS, WRAT and Wechsler Objective Reading Dimensions (WORD). Performance IQ in GenLang was based on subtests of IQ tests that did not depend on verbal cues, as included for example in the BAS and Wechsler Intelligence Scale for Children (WISC). Trait descriptions and summary statistic sources are in Supplementary Table 22. Bonferroni correction for multiple testing derived an adjusted critical $P$ value of $5.1 \times 10^{-4}$ from 98 independent tests.

Genetic correlations were further estimated in a targeted analysis of structural brain magnetic resonance imaging measures from UK Biobank, which were more comprehensive than those currently available from ENIGMA, along with further advantages such as hemisphere-specific data and greater homogeneity in cohort and scanning procedures. GWAS summary statistics from brain imaging-derived phenotypes for 33,000 participants were downloaded from the Oxford Brain Imaging Genetics Server[84]. Structural brain imaging traits encompassed both diffusion tensor imaging and surface-based morphometric phenotypes[85] where selected tracts or regions of interest had a known link to language. For diffusion tensor imaging, fractional anisotropy values derived from both tract-based-spatial statistics and probabilistic tractography were used for available tracts spanning the extended language network[86]. For surface-based morphometric (cortical volume, surface area and thickness) GWAS, summary statistics for regions of interest derived from the Desikan-Killiany atlas (white surface) were used, again selected for their relevance in language processing, based on previous literature[87–90]. To correct for multiple testing, phenotypic correlations between the UK Biobank imaging indices were derived and analyzed by PhenoSpD[23] to obtain the number of independent variables (36.08) to use for Bonferroni correction (adjusted critical $P$ value of $1.39 \times 10^{-3}$).

## Polygenic score analyses

Dyslexia polygenic scores were based on increasingly larger numbers of SNPs corresponding to their association $P$ values from the 23andMe GWAS ($P < 5 \times 10^{-8}$, $P < 1 \times 10^{-5}$, $P < 0.001$, $P < 0.01$, $P < 0.05$, $P < 0.1$, $P < 0.5$, 1). They were calculated in four independent cohorts. Two were general population cohorts from Australia: $n = 1,640$ (772 families) adolescents/young adults (Brisbane adolescents)[91]; $n = 1,165$ (966 families) older adults (Brisbane adults)[25]. The other two were family-based samples selected for dyslexia: one from the United Kingdom (UKdys), $n = 930$ (595 families); the other from the United States (Colorado Learning Disabilities Research Center, CLDRC), $n = 717$ (336 families)[92]. In the Australian samples, polygenic scores were calculated on 1000 Genomes Phase 3 (v.20101123) imputed genetic data using PLINK[93]. Only reliably imputed SNPs ($R^2 > 0.80$) and those with a minor allele frequency >0.01 were included, and the default clumping procedure was used where index SNPs formed a clump with other SNPs in LD ($R^2 > 0.1$) and within a 250 kb distance. In the UKdys and CLDRC samples, polygenic scores were calculated on Haplotype Reference Consortium imputed genetic data using PRSice[94], with the same imputation quality and MAF exclusions for the base (23andMe GWAS) sample, and clumping parameters.

Polygenic scores were then used as predictors in linear models of quantitative trait outcomes (Australia: word, nonword (phonetic), irregular word (lexical) reading and spelling tests from an extended version of the Components of Reading Examination[95], and two non-word repetition tests which are sensitive to developmental language disorders—Dollaghan and Campbell[96], Gathercole and Baddeley[97]; UKdys and CLDRC: word recognition). All quantitative traits were pre-adjusted for sex, age and ancestry principal components (10 principal components in UKdys and CLDR; 20 principal components in Australian samples). Further adjustments were made for imputation run (separate runs for different genotyping arrays) in the Australian samples, and for nonverbal IQ in all samples (except for the Australian adults), and for hearing difficulties in the Australian older adults. Because the cohorts included related family members (twins or siblings), linear mixed models (lme) were specified in RStudio[98], with family membership modeled as a random effect and the dyslexia polygenic score as a fixed effect. Where monozygotic twins were present, their trait scores were averaged and they were used as a single case.

### Evaluation of candidates from previous literature

We used the results of the 23andMe dyslexia GWAS to assess variants, genes and biological pathways previously associated with or implicated in dyslexia and/or variation in reading and spelling ability in past association studies, linkage analyses and other studies.

**Previously reported variants.** We assessed 75 previously reported variants within our summary statistics, adopting a replication/validation significance threshold of $P < 7.28 \times 10^{-4}$, derived by Bonferroni correction based on 68.7 independent tests derived through matrix spectral decomposition, taking into account LD (see Doust et al.[25] for details on how these variants were selected). The sources for each variant are provided in Supplementary Table 26.

**Dyslexia candidate genes.** We evaluated gene-based results from MAGMA v.1.08 (ref. [56]) for overrepresentation of genome-wide significant variants from the 23andMe dyslexia GWAS within the loci of 14 candidate genes from earlier literature: *CMIP*, *CNTNAP2*, *CYP19A1*, *DCDC2*, *DIP2A*, *DYX1C1*, *GCFC2*, *KIAA0319*, *KIAA0319L*, *MRPL19*, *PCNT*, *PRMT2*, *S100B* and *ROBO1*. The rationale for this selection is detailed by Luciano et al.[24] and Doust et al.[5]. The critical P value, based on Bonferroni correction for 14 tests, was $3.57 \times 10^{-3}$.

**Candidate dyslexia gene sets.** We performed a gene set analysis in MAGMA to test for overrepresentation of genome-wide significant variants within (1) a set of transcriptional targets of *FOXP2*, a highly conserved transcription factor linked to speech and language impairment[99]; and (2) two biological pathways previously suggested to play a role in dyslexia susceptibility[100,101]—axon guidance (GO:0007411: 'chemotaxis process that directs the migration of an axon growth cone to a specific target site'; 216 genes) and neuron migration (GO:0001764: 'movement of an immature neuron from germinal zones to specific positions where they will reside as they mature'; 145 genes). An adjusted critical P value of 0.017 was derived using Bonferroni correction based on three independent tests.

### Ethical standards

Participants provided informed consent and participated in the research online, under a protocol approved by the external AAHRPP-accredited IRB, Ethical and Independent Review Services. Participants were included in the analysis on the basis of consent status as checked at the time data analyses were initiated.

### Reporting summary

Further information on research design is available in the Nature Research Reporting Summary linked to this article.

### Data availability

The full summary statistics for each dyslexia GWAS presented in this paper will be made available through 23andMe website (https://research.23andme.com/dataset-access/) to qualified researchers under an agreement with 23andMe that protects the privacy of the 23andMe participants. The top 10,000 associated SNPs from the main GWAS can be downloaded from https://doi.org/10.7488/ds/3465.

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

## Acknowledgements

We thank the research participants and employees of 23andMe Inc, the GenLang Consortium, the Brisbane Adults Reading Study, and the CRS. E.E., G.A., B.M., B.S.P., C.F. and S.E.F. are supported by the Max Planck Society (Germany). The CRS was supported by grants from the National Natural Science Foundation of China (Grant No. 61807023), Funds for Humanities and Social Sciences Research of the Ministry of Education (Grant No. 19YJC190023 and 17XJC190010) and General Project of Shaanxi Natural Science Basic Research Program (2018JQ8015) (Grant No. 2018JQ8015 and 2021JQ-309). S.P. is funded by the Royal Society. Acknowledgements for the GenLang Consortium appear in the Supplementary Note.

## Author contributions

M.L., S.E.F., T.C.B. and N.G.M. conceived the study, with M.L. overseeing general analysis and A.A. overseeing 23andMe analysis. C.D., P.F., E.E., G.A., S.D.G., Z.W., B.M. and M.L. performed statistical and/or downstream annotation analysis. R.E.M. advised C.D. on some analysis. C.D. drafted the manuscript, with sections contributed by P.F., E.E., G.A., Z.W. and M.L. B.S.P., C.F. and S.E.F. supervised the GenLang GWAS. J.Z. managed the Chinese Reading Study. S.P., J.B.T., A.P.M. and J.F.S. managed the UKDys study. J.R.G., R.K.O., E.G.W., J.C.D., B.F.P. and S.D.S. managed the CLDRC study. M.J.W., T.C.B. and N.G.M. managed the Australian adolescent twin studies. M.L., T.C.B., S.E.F. and N.G.M. managed the Australian adult reading study. All authors critically reviewed the manuscript.

## Competing interests

P.F., A.A. and the 23andMe Research Team are employed by and hold stock or stock options in 23andMe, Inc. The remaining authors declare no competing interests.

## Additional information

**Extended data** is available for this paper at https://doi.org/10.1038/s41588-022-01192-y.

**Correspondence and requests for materials** should be addressed to Michelle Luciano.

All

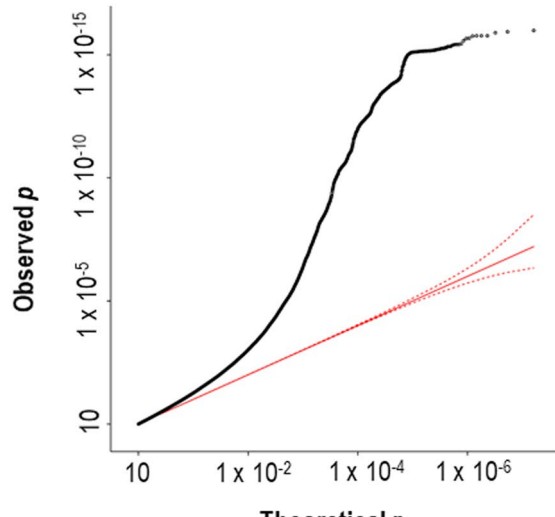

Female

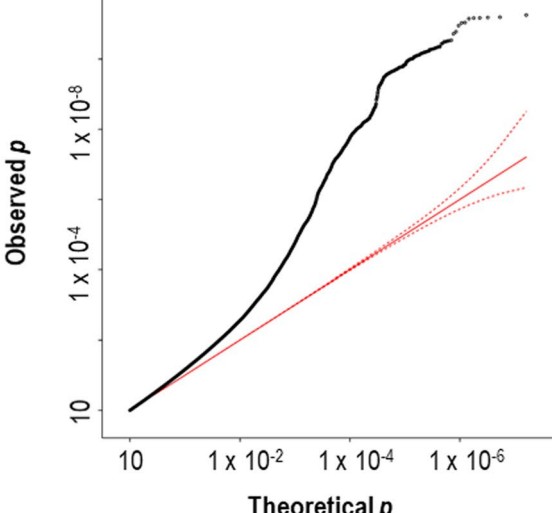

Male

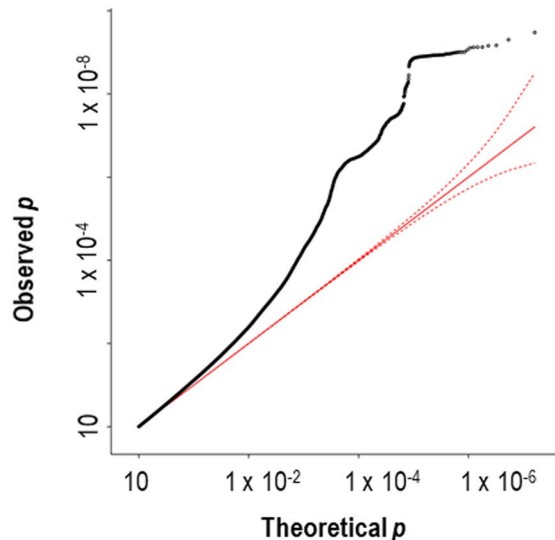

**Extended Data Fig. 1 | See next page for caption.**

**Extended Data Fig. 1 | QQ plot of dyslexia GWAS results. a-c**, Quantile-quantile (Q-Q) plots of observed versus expected *P* values for associations of single nucleotide polymorphisms with self-reported dyslexia diagnosis in a genome-wide association analysis for all participants (*n* = 51,800 cases, 1,087,070 controls) (**a**), female participants (*n* = 30,287 cases, 641,016 controls) (**b**), and male participants (*n* = 21,513 cases, 446,054 controls) (**c**). The solid red line represents the distribution of *P* values under the null hypothesis, and the dashed red line represent 95% confidence intervals. The black circles represent the observed distribution of *P* values.

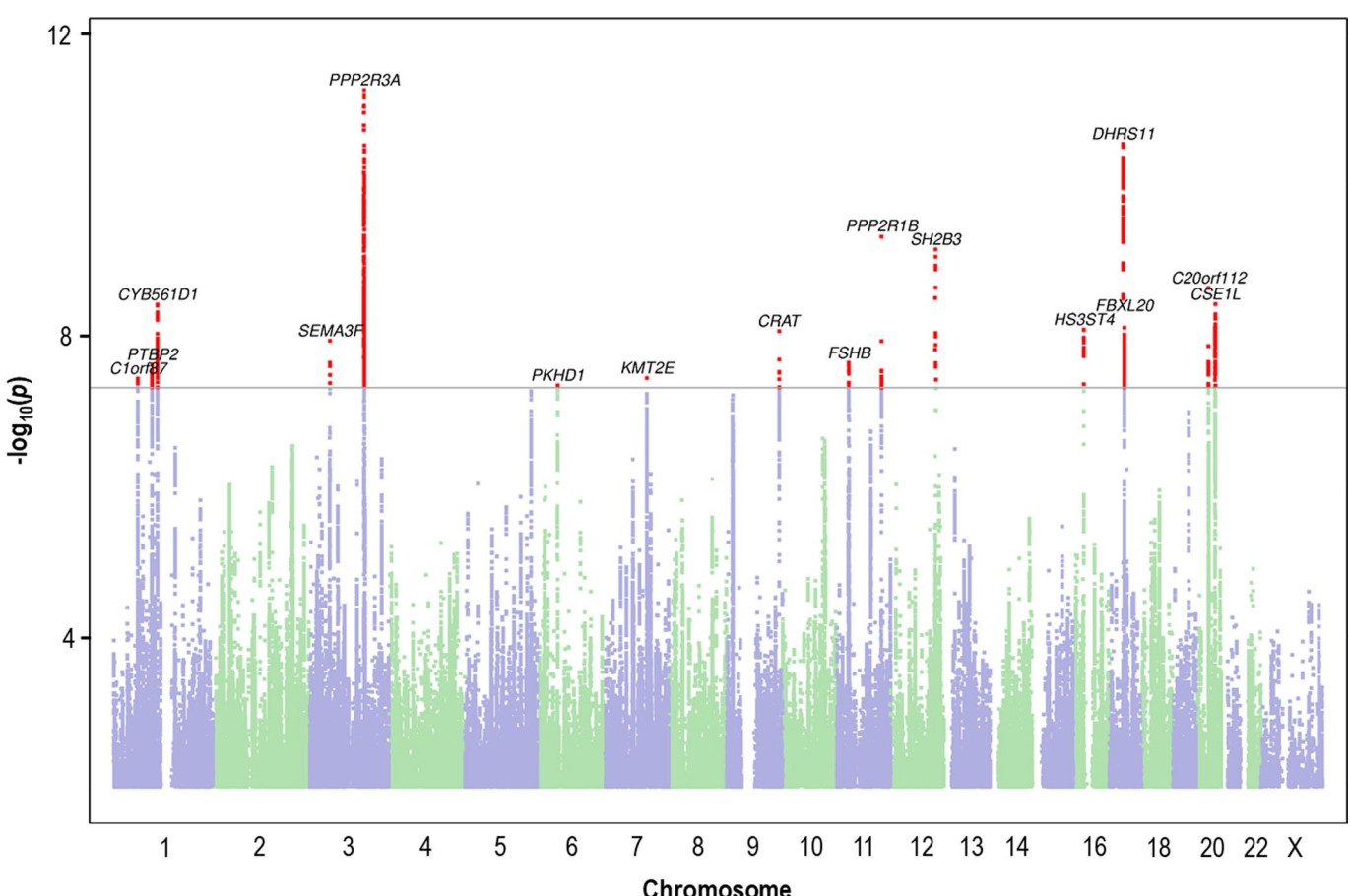

**Extended Data Fig. 2 | Manhattan plot of dyslexia GWAS results for females.**
The $y$-axis represents the -$\log_{10} P$ value for association of single nucleotide polymorphisms with self-reported dyslexia diagnosis from 30,287 female individuals and 641,016 female controls. The threshold for genome-wide significance ($P < 5 \times 10^{-8}$) is represented by a horizontal grey line. Genome-wide significant variants in the 17 genome-wide significant loci are red. Variants located within a distance of 250 kb of each other are considered as one locus.

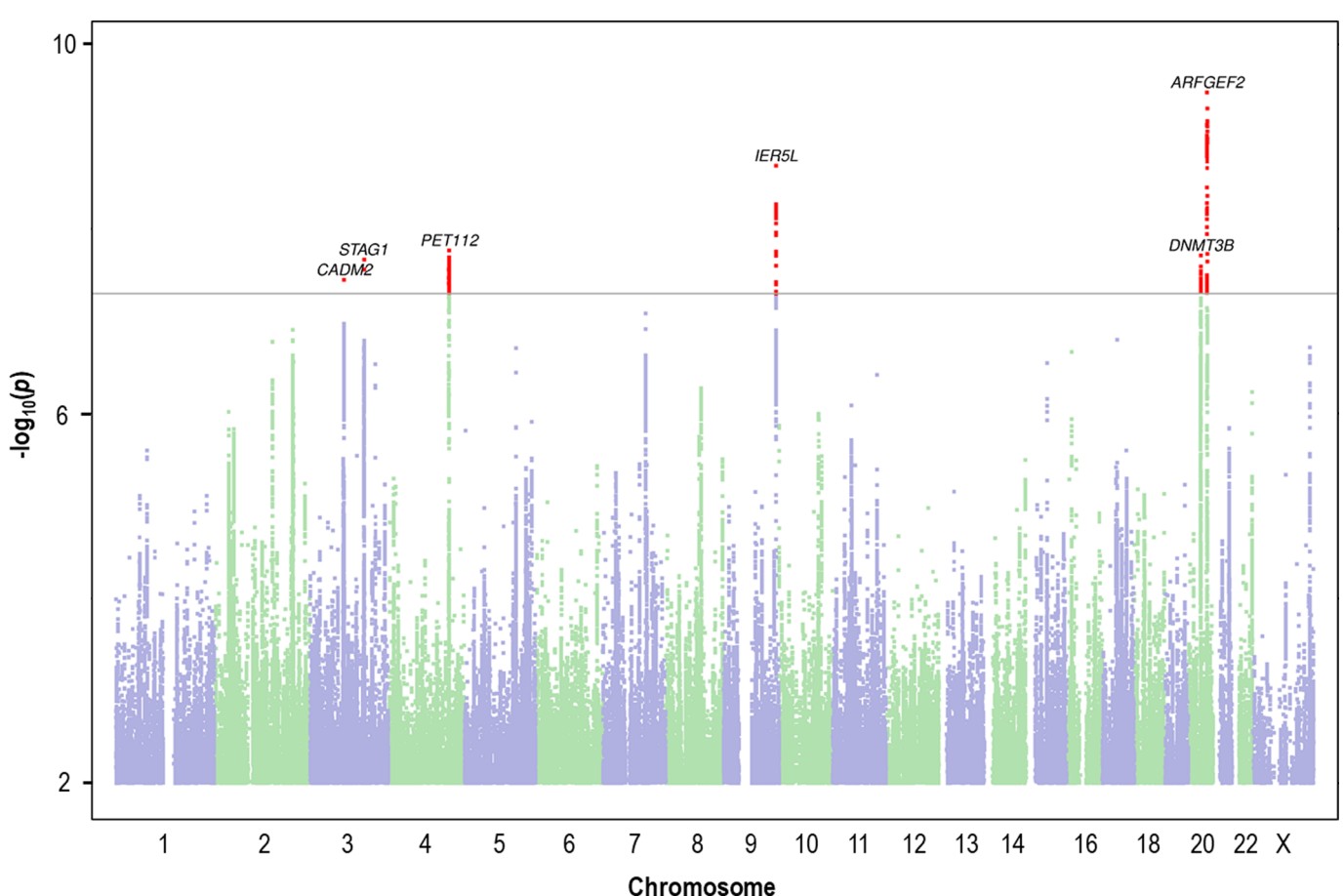

**Extended Data Fig. 3 | Manhattan plot of dyslexia GWAS results for males.** The $y$-axis represents the $-\log_{10} P$ value for association of single nucleotide polymorphisms with self-reported dyslexia diagnosis from 21,513 male individuals and 446,054 male controls. The threshold for genome-wide significance ($P < 5 \times 10^{-8}$) is represented by a horizontal grey line. Genome-wide significant variants in the 6 genome-wide significant loci are red. Variants located within a distance of 250 kb of each other are considered as one locus.

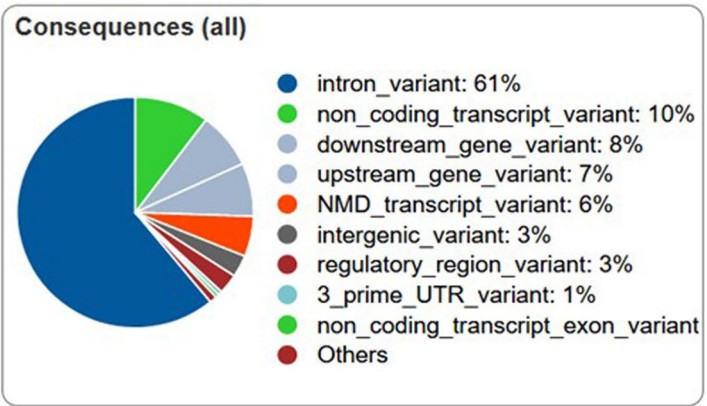

| Category | Count |
|---|---|
| Variants processed | 6210 |
| Variants filtered out | 0 |
| Novel / existing variants | 0 (0.0) / 6210 (100.0) |
| Overlapped genes | 238 |
| Overlapped transcripts | 1176 |
| Overlapped regulatory features | 569 |

**Consequences (all)**

- intron_variant: 61%
- non_coding_transcript_variant: 10%
- downstream_gene_variant: 8%
- upstream_gene_variant: 7%
- NMD_transcript_variant: 6%
- intergenic_variant: 3%
- regulatory_region_variant: 3%
- 3_prime_UTR_variant: 1%
- non_coding_transcript_exon_variant
- Others

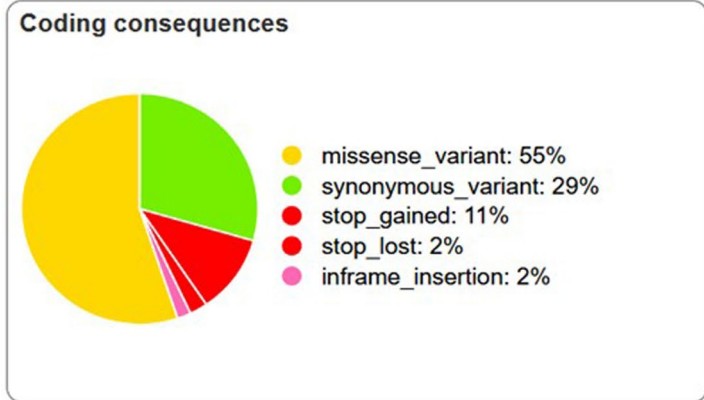

**Coding consequences**

- missense_variant: 55%
- synonymous_variant: 29%
- stop_gained: 11%
- stop_lost: 2%
- inframe_insertion: 2%

**Extended Data Fig. 4 | Variant effect predictor summary for the credible set of variants significantly associated with dyslexia.** Summary information is output from the online variant effect predictor in ENSEMBL (release 104). All our variants were present in the 1000 Genomes reference panel so are considered existing, and no pre-filtering (for example, on MAF; consequence type) was done.

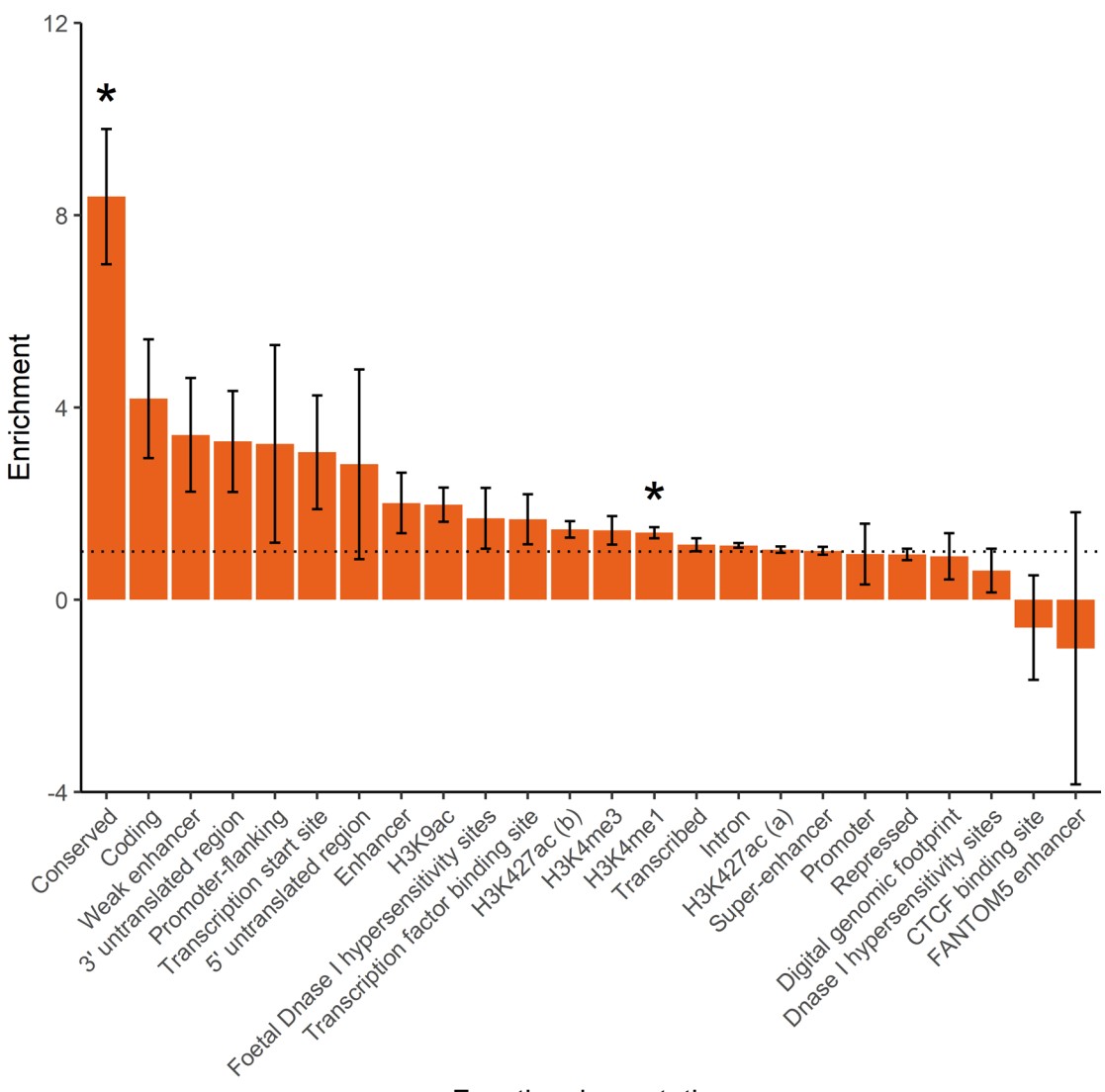

**Extended Data Fig. 5 | Enrichment estimates for major functional annotations.** The 24 major functional annotations were defined by Finucane et al.[39]. Enrichment is the proportion of $h^2$/proportion of SNPs. The horizontal dotted line indicates no enrichment (where proportion of $h^2$/proportion of SNPs = 1). Error bars represent standard errors of the enrichment estimates. Asterisks indicate enrichment estimates are significant based on a Bonferroni-derived $P$ value of $< 2.08 \times 10^{-3}$ (for 24 tests). Exact values of enrichment statistic, standard error, and $P$ value can be found in Supplementary Table 16.

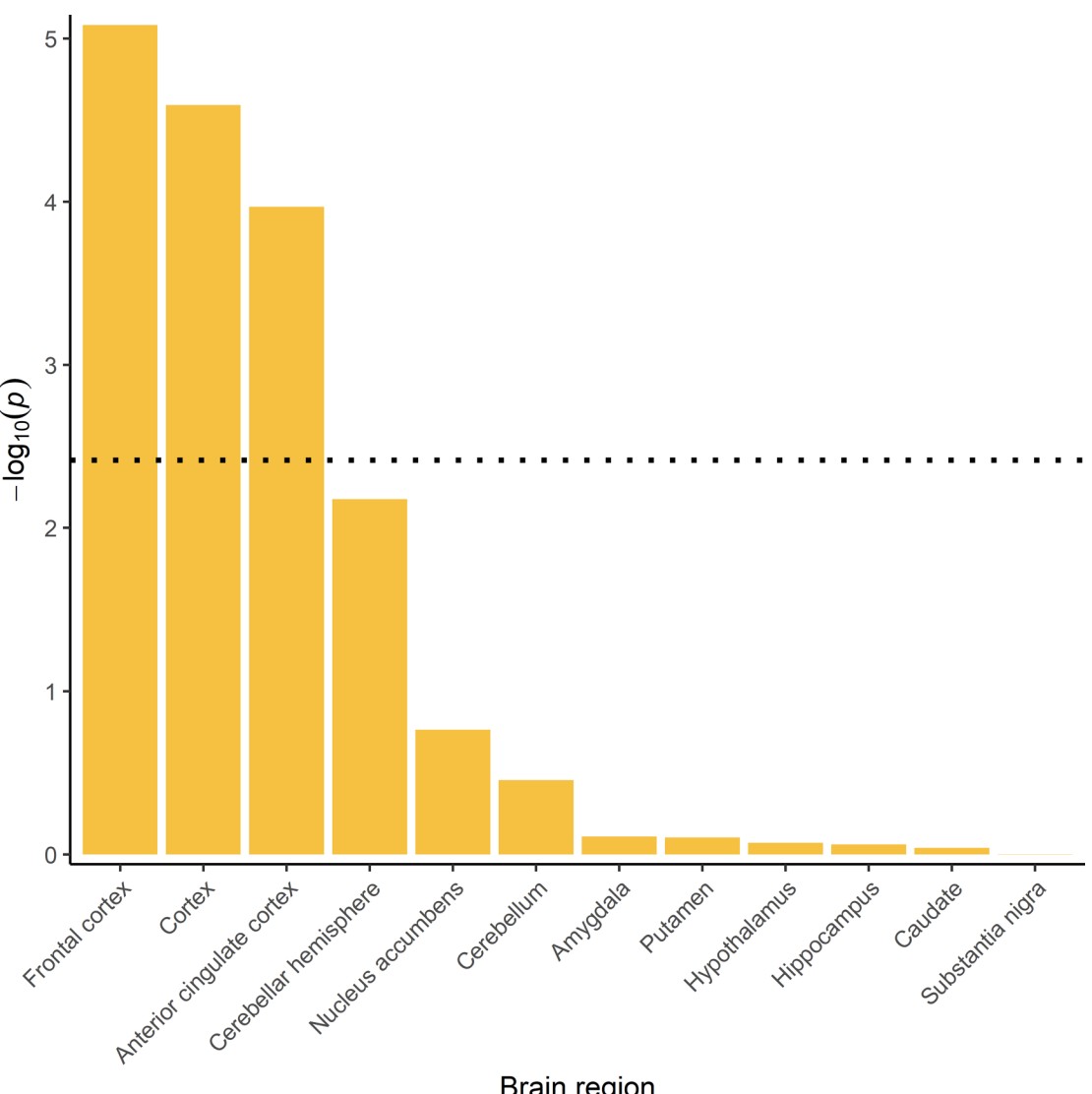

**Extended Data Fig. 6 | Heritability of dyslexia partitioned by brain tissue gene expression.** The -log$_{10}$ $P$ value of the enrichment estimates for heritability of dyslexia for genes expressed in 12 brain regions. The horizontal dotted line indicates significance after Bonferroni correction for 12 tests ($P < 4.17 \times 10^{-3}$).

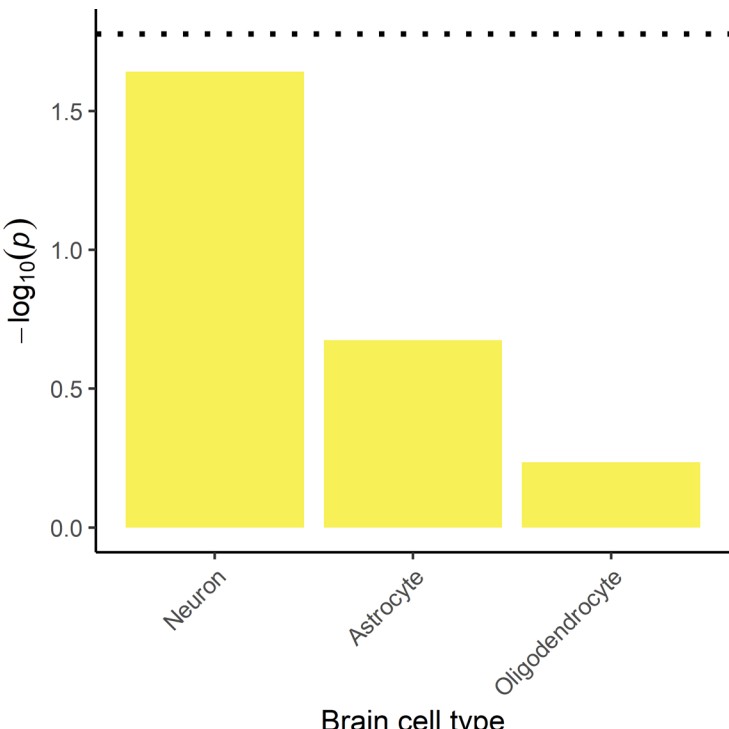

**Extended Data Fig. 7 | Heritability of dyslexia partitioned by brain cell type.** The -$\log_{10} P$ value of the enrichment estimates for heritability of dyslexia for brain cell types. The horizontal dotted line indicates significance after Bonferroni correction for three tests ($P < 1.67 \times 10^{-2}$).

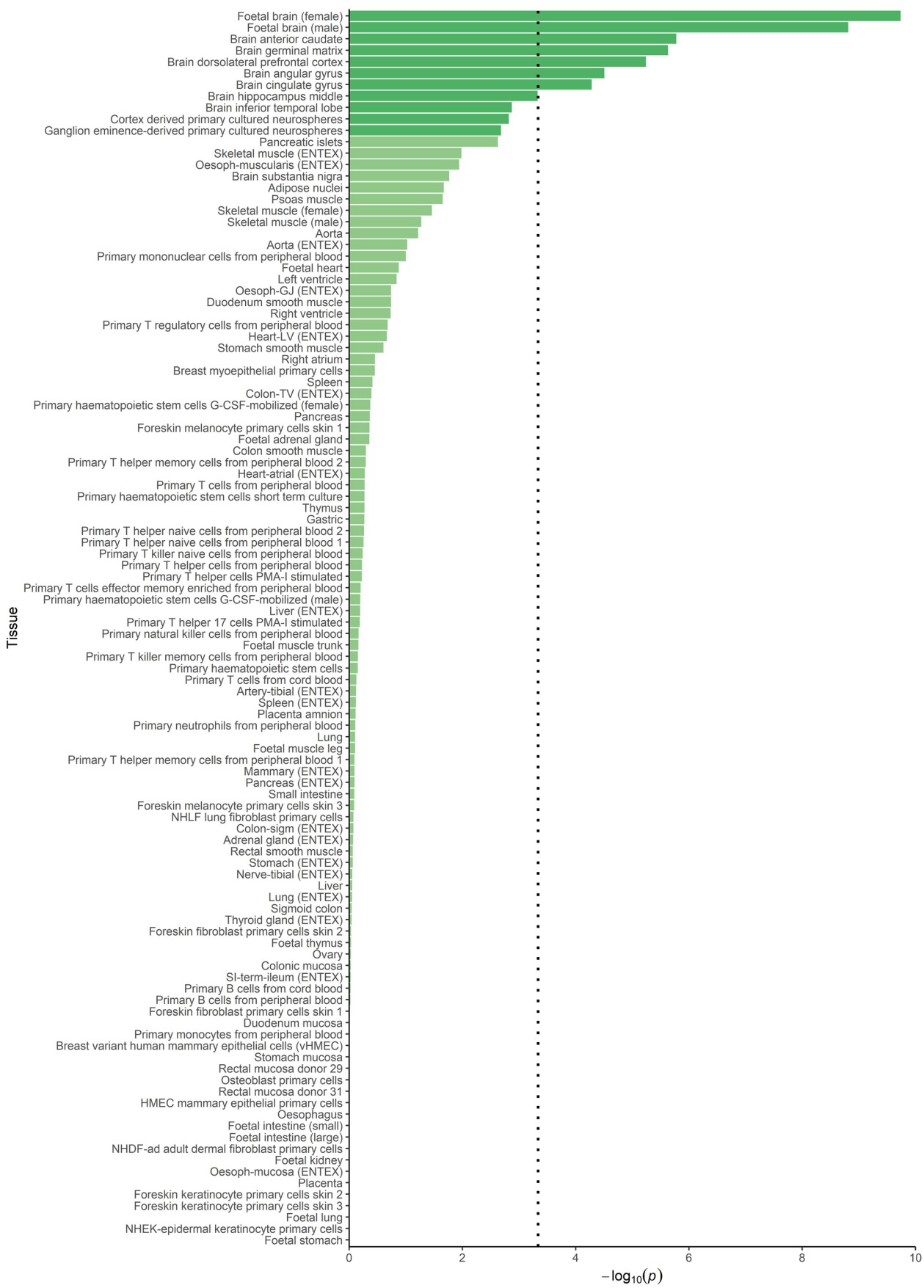

**Extended Data Fig. 8 | Heritability of dyslexia partitioned by cell-type specific H3K4me1.** The -log$_{10}$ P value of the enrichment estimates for heritability of dyslexia for variants located within H3K4me1 peaks of different tissues. Central nervous systems tissues are represented in dark green and other tissues are represented in light green. The vertical dotted line indicates significance after Bonferroni correction for 114 tests ($P < 4.39 \times 10^{-4}$).

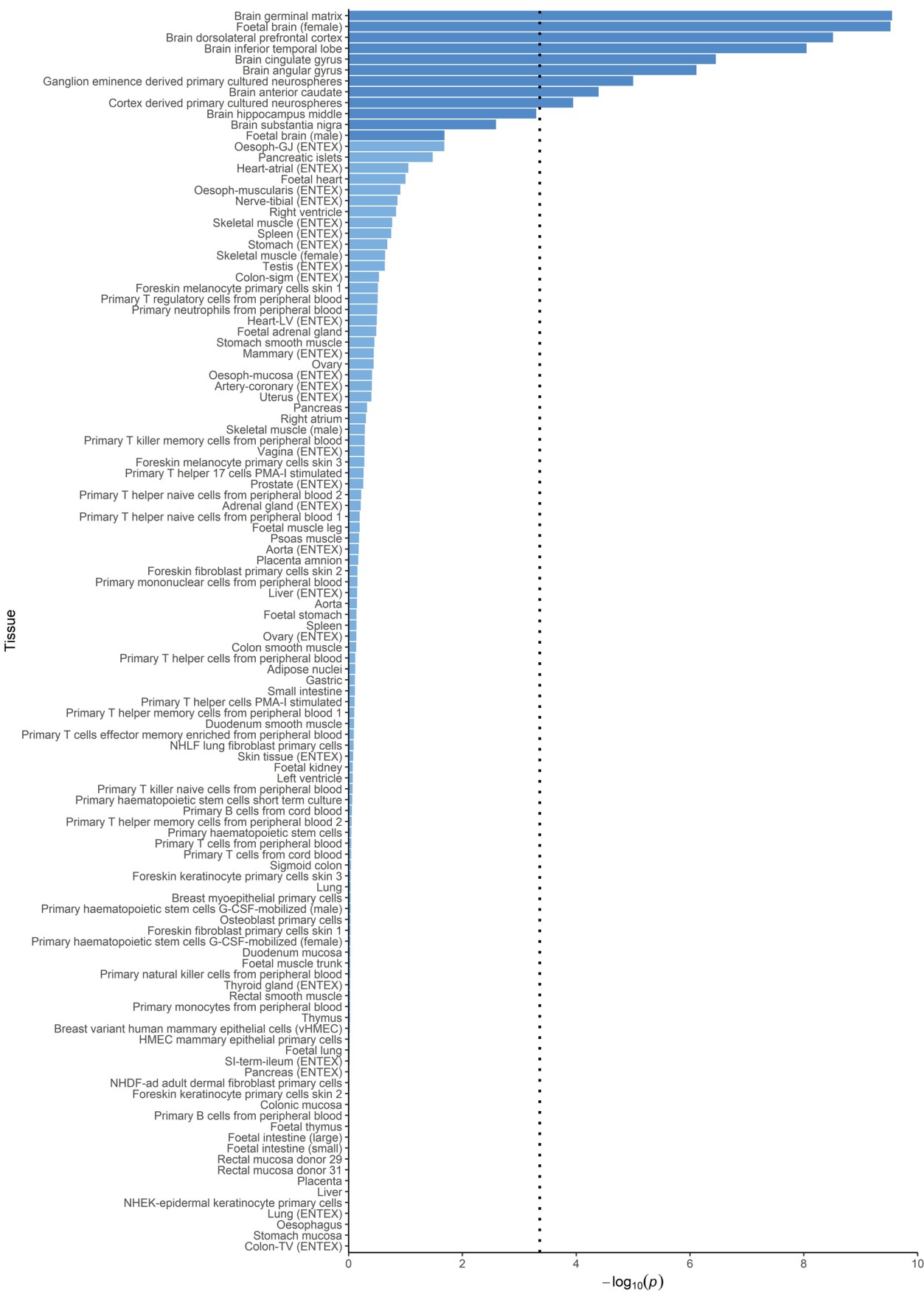

**Extended Data Fig. 9 | Heritability of dyslexia partitioned by cell-type specific H3K4me3.** The -log₁₀ P value of the enrichment estimates for heritability of dyslexia for variants located within H3K4me3 peaks of different tissues. Central nervous systems tissues are represented in dark blue and other tissues are represented in light blue. The vertical dotted line indicates significance after Bonferroni correction for 114 tests ($P < 4.39 \times 10^{-4}$).

# Reporting Summary

## Statistics

For all statistical analyses, confirm that the following items are present in the figure legend, table legend, main text, or Methods section.

| n/a | Confirmed | |
|---|---|---|
| ☐ | ☒ | The exact sample size ($n$) for each experimental group/condition, given as a discrete number and unit of measurement |
| ☐ | ☒ | A statement on whether measurements were taken from distinct samples or whether the same sample was measured repeatedly |
| ☐ | ☒ | The statistical test(s) used AND whether they are one- or two-sided<br>*Only common tests should be described solely by name; describe more complex techniques in the Methods section.* |
| ☐ | ☒ | A description of all covariates tested |
| ☐ | ☒ | A description of any assumptions or corrections, such as tests of normality and adjustment for multiple comparisons |
| ☐ | ☒ | A full description of the statistical parameters including central tendency (e.g. means) or other basic estimates (e.g. regression coefficient) AND variation (e.g. standard deviation) or associated estimates of uncertainty (e.g. confidence intervals) |
| ☐ | ☒ | For null hypothesis testing, the test statistic (e.g. $F$, $t$, $r$) with confidence intervals, effect sizes, degrees of freedom and $P$ value noted<br>*Give P values as exact values whenever suitable.* |
| ☒ | ☐ | For Bayesian analysis, information on the choice of priors and Markov chain Monte Carlo settings |
| ☒ | ☐ | For hierarchical and complex designs, identification of the appropriate level for tests and full reporting of outcomes |
| ☐ | ☒ | Estimates of effect sizes (e.g. Cohen's $d$, Pearson's $r$), indicating how they were calculated |

*Our web collection on statistics for biologists contains articles on many of the points above.*

## Software and code

Policy information about availability of computer code

| Data collection | 23andMe online survey software. |
|---|---|
| Data analysis | MAGMA v1.08; FUMA v1.3.6a; LD Hub v1.9.3; LDSC (LD SCore) v1.0.1; R 4.0.0;  Minimac 3 V2.0.1; PLINK 2.0<br>No custom code was written. |

For manuscripts utilizing custom algorithms or software that are central to the research but not yet described in published literature, software must be made available to editors and reviewers. We strongly encourage code deposition in a community repository (e.g. GitHub). See the Nature Portfolio guidelines for submitting code & software for further information.

## Data

Policy information about availability of data

All manuscripts must include a data availability statement. This statement should provide the following information, where applicable:
- Accession codes, unique identifiers, or web links for publicly available datasets
- A description of any restrictions on data availability
- For clinical datasets or third party data, please ensure that the statement adheres to our policy

The full summary statistics for each dyslexia GWAS presented in this paper will be made available through 23andMe to qualified researchers under an agreement with 23andMe that protects the privacy of the 23andMe participants. Interested investigators should email dataset-request@23andme.com and reference this paper for more information.

# Field-specific reporting

Please select the one below that is the best fit for your research. If you are not sure, read the appropriate sections before making your selection.

☐ Life sciences  ☒ Behavioural & social sciences  ☐ Ecological, evolutionary & environmental sciences

For a reference copy of the document with all sections, see nature.com/documents/nr-reporting-summary-flat.pdf

# Behavioural & social sciences study design

All studies must disclose on these points even when the disclosure is negative.

| | |
|---|---|
| Study description | This is a quantitative study that relies on self-report data. Replication analyses draw on cognitive test data. |
| Research sample | The research sample are customers of 23andMe, a consumer genetics company, who have agreed to participate in research. They are slightly selected in that there is over-representation of higher socio-economic position participants. The replication samples are from the general population and some are enriched from reading difficulties. |
| Sampling strategy | The sample were volunteer customers of 23andMe, who consented to the use of their DNA and survey results. The largest sample size available at the time of the study were used. |
| Data collection | Data collection for the main analysis was online survey collection. For replication samples it was mostly in-person cognitive testing. |
| Timing | The main sample data include customers who consented to participate up until late 2020. The replication sample cohorts vary in data collection times with some dating back to the 90s. |
| Data exclusions | No exclusions were made. |
| Non-participation | This is not a longitudinal study so there is no sample drop-out to report. |
| Randomization | There was no randomization but we controlled for age and sex in the analyses. |

# Reporting for specific materials, systems and methods

We require information from authors about some types of materials, experimental systems and methods used in many studies. Here, indicate whether each material, system or method listed is relevant to your study. If you are not sure if a list item applies to your research, read the appropriate section before selecting a response.

## Materials & experimental systems

| n/a | Involved in the study |
|---|---|
| ☒ | ☐ Antibodies |
| ☒ | ☐ Eukaryotic cell lines |
| ☒ | ☐ Palaeontology and archaeology |
| ☒ | ☐ Animals and other organisms |
| ☐ | ☒ Human research participants |
| ☒ | ☐ Clinical data |
| ☒ | ☐ Dual use research of concern |

## Methods

| n/a | Involved in the study |
|---|---|
| ☒ | ☐ ChIP-seq |
| ☒ | ☐ Flow cytometry |
| ☒ | ☐ MRI-based neuroimaging |

## Human research participants

Policy information about studies involving human research participants

| | |
|---|---|
| Population characteristics | See above. |
| Recruitment | Participants are customers of 23andMe, so are invited to participate in general research, our study uses data from a number of online questions. There is under-representation of low socio-economic position and all participants are over 18 years. |
| Ethics oversight | External AAHRPP-accredited IRB, Ethical & Independent Review Services. |

Note that full information on the approval of the study protocol must also be provided in the manuscript.

