## [Peer Review File · Nature Genetics]

Peer Review Information

Manuscript Title: Discovery of 42 genome-wide significant loci associated with dyslexia

Corresponding author name(s): Michelle Luciano

Reviewer Comments & Decisions:

Decision Letter, initial version:
--

27th September 2021

Dear Michelle,

Your Article "Discovery of 42 genome-wide significant loci associated with dyslexia" has been seen by three referees. You will see from their comments below that, while they find your work of interest, they have raised several relevant points. We are interested in the possibility of publishing your study in Nature Genetics, but we would like to consider your response to these points in the form of a revised manuscript before we make a final decision on publication.

To guide the scope of the revisions, the editors discuss the referee reports in detail within the team, including with the chief editor, with a view to identifying key priorities that should be addressed in revision, and sometimes overruling referee requests that are deemed beyond the scope of the current study. In this case, we ask that you address all technical points related to the analyses and their interpretation and revise the presentation throughout taking these points into account. In particular, we ask that you adjust the significance thresholds where needed as requested by the referees, and we also recommend that you remove the Mendelian randomization analyses for the reasons noted by the referees. We hope you will find this prioritized set of referee points to be useful when revising your study. Please do not hesitate to get in touch if you would like to discuss these issues further.

We therefore invite you to revise your manuscript taking into account all reviewer and editor comments. Please highlight all changes in the manuscript text file. At this stage, we will need you to upload a copy of the manuscript in MS Word .docx or similar editable format.

*2) If you have not done so already please begin to revise your manuscript so that it conforms to our Article format instructions, available [here](http://www.nature.com/ng/authors/article_types/index.html). Refer also to any guidelines provided in this letter.

[REDACTED]

We hope to receive your revised manuscript within 8-12 weeks. If you cannot send it within this time, please let us know.

Please do not hesitate to contact me if you have any questions or would like to discuss these revisions

further.

Sincerely,
Kyle

Kyle Vogan, PhD
Senior Editor
Nature Genetics
<https://orcid.org/0000-0001-9565-9665>

Referee expertise:

Referee #1: Genetics, complex traits, statistical methods

Referee #2: Genetics, behavioral traits, statistical methods

Referee #3: Genetics, complex traits, dyslexia

Reviewers' Comments:

Reviewer #1:

3Remarks to the Author:

This is a well-done genome-wide association study (GWAS) of dyslexia. Using data from 23andMe, the authors are able to conduct this study in a much larger sample than has been possible previously. In this paper, the authors identify 42 independent genome-wide significant loci, including several that have not been previously associated with other cognitive traits. They also perform a number of follow-up analyses (including genetic correlations estimation, heritability partitioning, and Mendelian randomization) to understand the genetics of dyslexia.

Overall, I thought the paper felt complete and well written. I was largely able to follow what the authors were doing and how they justified their analyses and conclusions. I especially appreciated the authors' efforts to include more than just European-ancestry data in their study. This study will likely be broadly useful to the research community.

There were a handful of areas, however, that left me with questions. I describe these below.

Major Concerns

1) I found the Mendelian randomization section to be quite weak, especially relative to the high quality work in the rest of the paper. There was virtually no discussion of the assumptions required to interpret the results other than acknowledging that dynastic, assortative mating, and population stratification effects may bias the results. While I understand that the authors don't have the data available to formally test whether these factors are present in their applications, it seems important to at least point to what evidence is already available in the literature. For example, it is well known that all three of these biases are substantial for educational attainment (Brumpton et al. 2020). Given the close link between education and dyslexia, my prior is that these biases may also be important for a MR study of dyslexia. This would be especially true when the outcome or exposure variable is related to education, as they are in several of the MR analyses in this paper. Given these concerns, I don't know that we can learn much from the MR studies in the paper. I would GREATLY weaken the causal language, provide a more nuanced discussion of why readers should or shouldn't believe the MR assumptions, and explain what we can pull from the MR studies if the assumptions are violated. Alternatively, the authors could remove the MR analyses from the paper entirely.

2) I was surprised that the authors only compared their results to Eising et al. (bioRxiv) which apparently has only 20-30k samples. (As a side note, I tried to find Eising et al. on bioRxiv and I couldn't. Is it not posted yet or am I just an incompetent googler?) Becker et al. (2021) included a GWAS of childhood

reading in almost 200k individuals in a 23andMe sample. Given that 23andMe is a coauthor on this paper, I presume they would be able to access those summary statistics.

3) I was confused by your multiple testing correction in the heritability partitioning analyses and in the genetic correlation analyses. It appears that the authors split these analyses into different groups and only corrected for multiple testing within each group. So in some cases, you only correct for 4-5 partitioning tests when you actually have conducted over 100 partitioning tests in the paper. While I think it is standard to correct for all the genetic correlation tests separately from the partitioning tests, splitting the tests within each of these groups seem unconventional. The authors should argue that their correction approach is either standard or why their case requires something non-standard. If they choose to continue to split their tests this way, they should also report how many of their results survive a more stringent test that accounts for all genetic correlation (or partitioning) tests.

4) More detail on how the sample was selected would be useful in the SI. I would not be able to replicate the authors results with the amount of detail provided. For example, what 31 reference populations were used? Is there a citation for the Hidden Markov Model they use to measure admixture? If it's an unpublished approach, the authors should provide details on how it works. For the 23andMe reference panel, which countries are used? Do you do any additional filtering beyond "four grandparents from the same country"? For example, I imagine that many African Americans have all four grandparents born in the US, but I suspect that you would not include those individuals in the reference sample.

Minor Concerns:

5) The authors claim in the intro that dyslexia affects 10% of the population. In their data, however, only 5% of the sample reports having dyslexia. Is this evidence of selection in the sample? If individuals with dyslexia are less likely to be part of the sample, could this lead to collider bias in the results?

6) The authors report a different number of independent variants in the MR analysis (51) than they report in the GWAS (42). I think this is because they use a different locus definition in each case. Is that true? If so, why?

7) Related to point 6, I wasn't able to find an explanation of the locus definition in the main text, methods section, or supplement. I did find some text in the note to Figure 1 ("Variants located within a distance of < 250 kb of each other are considered as one locus."), but it seems this information should also go in the text somewhere. If the text in the figure note is complete, this seems like an odd locus

5definition. In other GWASs that I'm aware of, the definition generally uses a wider window and often includes an r -squared criterion. How did the authors select this criterion?

8) In lines 118-119, the authors report several p -values. Are these p -values for a test against the null of $r_g=0$? Wouldn't it make more sense to report p -values for a test of $r_g=1$? Or better yet, perhaps it would be best to just report the 95% confidence intervals for these estimates so readers can get a sense of how precise the estimates are more generally. (Likewise, several genetic correlations are reported in lines 165-176, and it would be helpful if confidence intervals were reported there too.)

References:

Becker, J., Burik, C. A., Goldman, G., Wang, N., Jayashankar, H., Bennett, M., ... & Okbay, A. (2021). Resource profile and user guide of the Polygenic Index Repository. *Nature Human Behaviour*, 1-15.

Brumpton, B., Sanderson, E., Heilbron, K., Hartwig, F. P., Harrison, S., Vie, G. Å., ... & Davies, N. M. (2020). Avoiding dynastic, assortative mating, and population stratification biases in Mendelian randomization through within-family analyses. *Nature Communications*, 11(1), 1-13.

Reviewer #2:

Remarks to the Author:

This is a GWAS of dyslexia with ~50K cases and ~1M controls (all from 23andMe). Forty-two genome-wide significant hits were found. The study seems to be well done and will be of great interest to researchers interested in mental abilities and reading. The most intriguing finding is a moderate genetic correlation between dyslexia and ambidexterity.

Here are my comments.

p. 6, lines 92-93: "We identified 42 independent genome-wide significant associated loci". As far as I can tell, the paper never gives its definition of an independent genome-wide significant locus. There are many reasonable ones, and any one of these will do. But it has to be stated somewhere what was used.

p. 6, lines 100-101: "pleiotropic with general cognitive ability based on HEIDI-outlier analysis". HEIDI is a Mendelian-randomization tool. This is the first time that I have seen it used specifically to identify

pleiotropic SNPs. But I strongly recommend dropping this. Its performance depends on the statistical power of the two GWAS, which the authors seem not to consider. It requires the specification of a significance threshold (the default is, last I checked, $p < .01$), and the authors do not address whether this is optimal for their purpose. Also, the definition of pleiotropy effectively adopted by the use of HEIDI is a rather strange one; it means that the SNP is more associated with cognitive ability than expected on the basis of a model where cognitive ability affects dyslexia (according to Methods line 114). Is it absolutely necessary to exclude such SNPs from the downstream selection of likely causal genes?

p. 8, lines 138-140: "Gene property analysis for general tissues confirmed the importance of brain and specific regions (Supplementary Tables 13 and 14)". What does this mean? Does this correspond to Methods, lines 130-137?

p. 8, lines 140-141: "Levels of brain expression for the 136 genes proximal to the top single variants are shown in Supplementary Table 15." This is confusing for several reasons. First, the previous paragraph spoke of 42 loci and the most probable causal genes for each one. This suggests 42 genes, although a greater number (i.e., 136) is logically consistent with what is written. Second, the word "proximal" suggests that the 136 genes are selected on basis of mere proximity to the likely causal SNPs and not by the "Overall V2G score" mentioned in the previous paragraph. Is this in fact the case? Third, regardless of how the previous two points are resolved, it seems clear that the authors have shifted from the 173 MAGMA-prioritized genes mentioned in line 135 to some other set of genes. (Or are the 136 a subset?) Readers who do not care to go back and forth between the main text and the supplementary material might be either lost or made suspicious by what seems to be arbitrary changes in the analysis. If indeed different sets of genes are being employed in different analyses, I suggest that the authors instead select one method for picking out genes as the "main" method and other methods as "robustness checks", always run the main genes through the analyses (e.g., expression), and give the results of the "robustness" genes as asides or in the supplementary material. This might be a bit artificial post hoc, but it will at least help readers follow along.

p. 9, line 175: "spectral decomposition of matrices". What does this mean?

pp. 9-10, Mendelian randomisation: Although MR has become standard in GWAS papers, this is a huge mistake. MR is simply not a reliable method for causal inference. See the clear statement in Bowden et al. (2015), the paper introducing MR-Egger: "The InSIDE assumption would be violated if the pleiotropic effects act via a confounder of the exposure-outcome association." But this is precisely the case if there are heritable confounders or the causal direction is backward. If the authors want to do an MR "type" method, why not the ingenious LCV method of O'Connor and Price (2018)? I suspect that the LCV

method is not more popular because it often does not deliver the results desired. (Incidentally the simulations reported in this paper show that Egger, median, and mode methods all do poorly.) The authors might be interested in the CAUSE method of Morrison et al. (2020), which if I understand correctly can discern a true causal relationship in certain situations where LCV cannot. If the authors do not want to bother with new analyses, then I suggest entirely omitting attempts to put a causal order to two genetically correlated variables. In the past, however, editors and reviewers have not followed my suggestion about this matter. If that happens again, then I ask that the caveat in the Discussion ("the result is tentative given the potential for unmeasured confounding with the genetic instrument," p. 14, lines 267-268) be stated earlier in the Results and later in the Methods, loud and clear.

p. 12: "Consistent with studies of other neurodevelopmental disorders, partitioning of SNP-based heritability revealed enrichment in conserved regions". As far as I know, such enrichment has been found for **all** traits in GWAS, not just neurodevelopmental disorders. To imply that dyslexia falls into a special group with other neurodevelopmental disorders is quite misleading.

Methods, lines 103-104: This is more of a suggestion than a prescription. I hesitate to tell authors to do some other analysis than the one they did when the alternative analysis is better, but it is unclear how much better or whether it will lead to substantially different results. But I think that the Maller method of picking out likely causal SNPs is outdated because it assumes only one causal SNP in the locus and ignores functional information. A better method is that of Weissbrod et al. (2020), which seems very feasible for these authors because all of their data comes from a single cohort (23andMe) whose LD might be estimated by them.

Methods, Partitioned heritability: Only Finucane et al. (2015) is cited for stratified LD Score regression. But it seems that the authors used baseline annotations introduced after the publication of that paper (e.g., Gazal et al., 2017). To the extent that additional annotations coincide with published papers, I think those papers should be cited.

Methods, line 189: I am interested in the Eising preprint but cannot find it. I think that preprints should not be treated differently from published papers in the references; please provide sufficient information for a reader to find the preprint.

Methods, line 229: A paper by Hill et al. (2016) is given as authority for a genetic correlation between childhood and adult IQ of 0.71. But according to the paper, this has a standard error of 0.10. A later paper by Sniekers et al. (2017) gives the genetic correlation as 0.89 with a standard error of 0.08. True, the SE is not that much smaller, but it is probably small enough to exclude 0.71 from the 95% CI. Why

not cite both papers and give the range 0.7-0.9.

Methods, line 246: "PRSice". Again, a suggestion and not a prescription. Why not a method that uses LD to convert GWAS summary statistics into partial regression coefficients (e.g., Prive et al., 2020). The resulting prediction R^2 will be at least a little better.

Gazal, S. et al. (2017). Linkage disequilibrium-dependent architecture of human complex traits shows action of negative selection. *Nature Genetics*, 49, 1421-1427.

Morrison, J., Knoblauch, N., Marcus, J.H., Stephens, M., & He, X. (2020). Mendelian randomization accounting for correlated and uncorrelated pleiotropic effects using genome-wide summary statistics. *Nature Genetics*, 52, 740-747.

O'Connor, L.J., & Price, A.L. (2018). Distinguishing genetic correlation from causation across 52 diseases and complex traits. *Nature Genetics*, 50, 1728-1734.

Prive, F. et al. (2020). LDpred2: better, faster, stronger. *Bioinformatics*, 36, 5424-5431.

Sniekers, S. et al. (2017). Genome-wide association meta-analysis of 78,308 individuals identifies new loci and genes influencing human intelligence. *Nature Genetics*, 49, 1107-1112.

Weissbrod, O. et al. (2020). Functionally informed fine-mapping and polygenic localization of complex trait heritability. *Nature Genetics*, 52, 1355-1363.

Reviewer #3:

Remarks to the Author:

The authors of this interesting study are to be commended on their undertaking of studying the genetics of dyslexia in their uniquely powered questionnaire-based dataset at 23andMe where 51,800 adult participants have answered the question: "Have you been diagnosed with dyslexia"? with a "yes" (cases) and 1,087,070 with a "no" (controls), giving a prevalence of 4.5% among responders. The low prevalence of dyslexia in this sample speaks to the probably high-functioning of diagnosed dyslexics who participate

9in 23andMe's studies. This case/control definition does not rule out, but may lower prevalence of undiagnosed dyslexia/reading difficulties from controls, and the authors identify 42 independent genome-wide loci, 17 of which are in genes harboring SNPs with previous associations with various cognitive traits or educational attainment. The other 25 loci are considered by the authors novel associations with dyslexia, as they have no previous GWAS associations with cognitive traits. This classification is debatable. However, the number of significant dyslexia associations, pleiotropic with cognitive traits or not, is a significant contribution to the genetics of dyslexia, of which little is previously known.

Overall, the GWAS methodology is well described and appropriate, although the conventional threshold $p < 5 \times 10^{-8}$ is somewhat relaxed for novel discovery. How many variants were tested?

The authors validate their results in samples (1) of 2,274 variously defined dyslexia cases from nine European countries (NeuroDys by Gialluisi et al.), (2) a population sample of 2,270 Chinese children measured on reading accuracy and reading fluency (Chinese Reading Study, CRS), and (3) word-reading and spelling traits (33,959 and 18,514 respectively) from GWAS meta-analysis of children and adolescents from Europe, USA and Australia (GenLang consortium) which is referenced under Eising et al. bioRxiv (line 106 p. 6). This reference for the largest replication cohort credited with the most replications in Table 1, was, however, not found in the manuscript references nor in bioRxiv.

Table 1 (in title: novel associations rather than novel variants) presents results for the 25 defined dyslexia loci from the discovery cohort. Effect allele is listed and frequency (presumably of effect allele?), but inconsistently presented is the risk allele or minor allele which is confusing. It would be better to select as effect allele, either always the minor allele, or always the risk allele. Validation cohort is presented in the last column with p-value only, but missing from the table are validation phenotype, size of validation sample and validation effects. Furthermore, the authors use $p < 0.05$ for replication significance. However, if 42 variants are tested for replication in 4 phenotypes (Supplementary Tables 3-6), the threshold for combined replication analysis requires further adjustment.

In Supplementary Tables 3 (GenLang reading ability – assume higher score is better?) and 4 (GenLang spelling ability – assume higher score better?), some SNPs are marked with *Consistent effect direction. Consistent with what?

The sex-specific GWAS is very interesting but results are only presented in Manhattan plots and genetic correlations. Authors state that sex-specific results show high consistency with the main GWAS. Yet the sex-specific Manhattan plots (Supplementary Figures 2 and 3) show that top associations for the sexes

are quite different with males driving some signals (e.g. ARFGEF2) and females others (e.g. PP2R3A). These sex differences merit more attention in the manuscript.

The genetic correlations with dyslexia depicted in Figure 2 and Supplementary Table 22 show results for 98 traits but unclear is how the authors selected these traits from the many more traits available in LD Hub? Traits of reading, spelling, intelligence, and handedness are justified by the authors, as are the phenotypes representing known psychiatric comorbidities. But why did the authors decide to select a set of pain phenotypes? As they state in the discussion, the correlation of dyslexia with pain was unexpected? The reference provided for pain correlation is [23] Bycroft, C., et al., The UK Biobank resource with deep phenotyping and genomic data. *Nature*, 372 2018. 562(7726): p. 203-209, indicating that many more phenotypes were available for testing.

Finally, under data availability the authors state that the "full summary statistics for each dyslexia GWAS presented in this paper will be made available through 23andMe to qualified researchers under an agreement with 23andMe that protects the privacy of the 23andMe participants". All research groups are equally mindful of privacy protection of their study participants, yet make summary statistics more readily available than 23andMe. Responsibly making data available to other research groups is the policy of Nature Genetics and other comparable journals and should apply to 23andMe as well.

Author Rebuttal to Initial comments

Dear Kyle,

We thank you for the opportunity to revise our manuscript in line with the very important feedback from the reviewers. We would like to point out first that, as requested by the editorial team, we have removed the Mendelian Randomisation (MR) results. Admittedly, we were unsure of whether to originally include them because of the difficulty in being able to satisfy some assumptions of the MR analysis, but thought they might suffice as exploratory findings. Given the implications, though, we agree that we should be cautious and exclude them. In our revision we have paid particular attention to all technical matters (detailed in response below), including adjustments to significance thresholds where needed, so that the analysis and interpretation is clearer. We hope you find the manuscript greatly improved. Note that we have also revised the manuscript in line with Nature Genetics formatting.

Response to Referees

Reviewer #1:

11Remarks to the Author: This is a well-done genome-wide association study (GWAS) of dyslexia. Overall, I thought the paper felt complete and well written. I was largely able to follow what the authors were doing and how they justified their analyses and conclusions. I especially appreciated the authors' efforts to include more than just European-ancestry data in their study. This study will likely be broadly useful to the research community. There were a handful of areas, however, that left me with questions. I describe these below.

Response: We thank you for your positive comments and seeing value in our study. We hope we have adequately answered your questions below.

Major Concerns

Comment 1) I found the Mendelian randomization section to be quite weak, especially relative to the high quality work in the rest of the paper. There was virtually no discussion of the assumptions required to interpret the results other than acknowledging that dynastic, assortative mating, and population stratification effects may bias the results. While I understand that the authors don't have the data available to formally tests whether these factors are present in their applications, it seems important to at least point to what evidence is already available in the literature. For example, it is well known that all three of these biases are substantial for educational attainment (Brumpton et al. 2020). Given the close link between education and dyslexia, my prior is that these biases may also be important for a MR study of dyslexia. This would be especially true when the outcome or exposure variable is related to education, as they are in several of the MR analyses in this paper. Given these concerns, I don't know that we can learn much from the MR studies in the paper. I would GREATLY weaken the causal language, provide a more nuance discussion of why readers should or shouldn't believe the MR assumptions, and explain what we can pull from the MR studies if the assumptions are violated. Alternatively, the authors could remove the MR analyses from the paper entirely.

Response: We share the same concerns as the reviewer and we spent much time deliberating over these findings and whether to include them. What was interesting among the results was that educational attainment results were generally not significant outcomes yet cognitive performance was; one might expect the former results to be more biased in the presence of dynastic effects and assortative mating. But like you say, we are unable to formally test for this bias. We viewed the lack of instruments for ADHD as the greatest limitation of these analyses; whereas there was support for dyslexia as a causal variable, our analyses could not rule out a bidirectional causal pathway from ADHD. It does seem premature to conclude that ADHD can be caused by dyslexia when this may not be the full story. Within-family studies are needed to address this, which we could plan to do in the future. As per the reviewer's and editors' request we have removed the MR analyses from the paper.

Comment 2) I was surprised that the authors only compared their results to Eising et al. (bioRxiv) which apparently has only 20-30k samples. (As a side note, I tried to find Eising et al. on bioRxiv and I couldn't. Is it not posted yet or am I just an incompetent googler?) Becker et al. (2021) included a

GWAS of childhood reading in almost 200k individuals in a 23andMe sample. Given that 23andMe is a coauthor on this paper, I presume they would be able to access those summary statistics.

Response: Firstly, we apologise for not updating the Eising et al., BioRxiv reference which unfortunately wasn't uploaded in time for inclusion. We should have replaced this placeholder with a published abstract. The pre-print is now available on BioRxiv, as follows:

Eising, E., et al. (2021). "Genome-wide association analyses of individual differences in quantitatively assessed reading- and language-related skills in up to 34,000 people." [bioRxiv: 2021.2011.2004.466897](https://doi.org/10.1101/2021.2011.2004.466897).

The 23andMe phenotype that is referred to as “childhood reading” by Becker et al. (2021) is not actually a measure of reading ability, but rather a single question about recollections of when a person estimates that they began reading (“at what age did you start to read?”), relying on self-report decades later. There are multiple important reasons why this phenotype cannot be taken as a proxy of reading ability. Firstly, we question the reliability of self-report data requiring recall of timing of specific events as far back as 3 years of age. This can be contrasted with questions about whether or not someone received a formal diagnosis of a learning disorder in their childhood, where we have little reason to doubt reliability of self-report. Indeed, adult self-reports of reading difficulties in childhood correlate well with adult reading achievement (e.g., Scarborough, 1984; Decker et al., 1989). Moreover, the meaning of “start to read” is open to a wide range of interpretations, from beginning to be able to recognize letters and words, through to becoming a proficient reader of sentences in books. But even if we were to assume that this were accurately recalled and that people are interpreting “start to read” in a consistent way, the age of learning to read is not a valid marker of reading proficiency because of the large variation in the developmental trajectories of reading acquisition; in other words, children who learn to read relatively late in childhood can still become perfectly proficient readers. Further, very early reading (at “under age 3”, one of the options for responding to the question) could reflect exceptionally high IQ/autistic special abilities rather than relating to reading skills per se, which introduces further noise into this self-report phenotype. Finally, given the wide age range of 23andMe participants, there will also be variation in “age started reading” simply due to cohort effects linked to systematic changes over time in education systems (i.e., at which age reading begins to be taught). While age appears to have been corrected for in the original GWAS, it is unclear what age-related variance is removed, since this will be a mixture of cohort effects and that due to age-related decline in memory which could affect self-report. Of note, practices for teaching literacy to children may not only differ over time, but also between countries and even within the same country (e.g. different states in the US) introducing further potential confounds in self-reports of “age started reading”.

For these reasons, we do not view this self-reported “age started reading” phenotype as a valid or reliable measure of reading skill or dyslexia, and thus doubt its relevance/utility for this particular study. We instead took advantage of the valid and reliable standardised tests of reading skills available in the GenLang consortium which, despite the smaller total sample size, represent gold standard phenotypes

based on direct quantitative assessments of abilities in participants, rather than self-report. The GenLang sample sizes may appear relatively modest in comparison to those obtainable via self-report approaches, but they are currently the largest of the field when it comes to these gold standard psychometric reading phenotypes, giving an opportunity to robustly validate our 23andMe dyslexia findings. Moreover, the GenLang cohorts are independent of the 23andMe sample, avoiding another potential source of bias. A future project could use genomicSEM methods to understand the multivariate relations between the 23andMe 'age started reading' self-report, general cognitive ability, and the GenLang standardised reading scores, adding clarity to the latent traits being measured, but this is clearly beyond the scope of the current paper.

Decker, S. N., Vogler, G. P., & Defries, J. C. (1989). Validity of self-reported reading disability by parents of reading-disabled and control children. *Reading and Writing*, 1(4), 327-331.

Scarborough, H.S. (1984). Continuity between childhood dyslexia and adult reading. *British Journal of Psychology*, 75: 329-348. <https://doi.org/10.1111/j.2044-8295.1984.tb01904.x>

Comment 3) I was confused by your multiple testing correction in the heritability partitioning analyses and in the genetic correlation analyses. It appears that the authors split these analyses into different groups and only corrected for multiple testing within each group. So in some cases, you only correct for 4-5 partitioning tests when you actually have conducted over 100 partitioning tests in the paper. While I think it is standard to correct for all the genetic correlation tests separately from the partitioning tests, splitting the tests within each of these groups seem unconventional. The authors should argue that their correction approach is either standard or why their case requires something non-standard. If they choose to continue to split their tests this way, they should also report how many of their results survive a more stringent test that accounts for all genetic correlation (or partitioning) tests.

Response: We decided on separate adjustment by different types of heritability partitioning analysis to avoid an overly stringent correction given that there will be overlap among annotations across the different series of analysis. Other research teams have implemented similar grouped Bonferroni corrections (e.g., Demontis et al, 2018) by functional, tissue group, and tissue specific analyses. However, in line with the suggestion to additionally report a more stringent correction, we define $p < 1.88 \times 10^{-4}$ (i.e., for 266 tests) as the global correction. This more conservative criterion only results in one test (H3K4me1 class) no longer being significant. We include this corrected value in our supplementary tables and add detail to the methods.

Inserted text in Methods of main document, page 17, 2nd sentence:

"Overall, we performed 266 different tests which would give a very conservative Bonferroni-corrected significance level of 1.88×10^{-4} , but because there will be overlap among annotation groups, we also report corrections to significance within different classes of annotation, each of which we now describe."

Comment 4) More detail on how the sample was selected would be useful in the SI. I would not be able to replicate the authors results with the amount of detail provided. For example, what 31 reference populations were used? Is there a citation for the Hidden Markov Model they use to measure admixture? If it's an unpublished approach, the authors should provide details on how it works. For the 23andMe reference panel, which countries are used? Do you do any additional filtering beyond "four grandparents from the same country"? For example, I imagine that many African Americans have all four grandparents born in the US, but I suspect that you would not include those individuals in the reference sample.

Response: Given that the selection methods have been previously reported by 23andMe, we preferred to be concise in our paper, with interested readers able to follow up the source material. But we inadvertently omitted important sources in our original submission, so we thank the reviewer for bringing this to our attention. We have now included a citation in the Supplementary Methods that fully details the methodology used for the ancestry analysis including how the reference populations were built, the Hidden Markov Model and the filtering. Please be aware that this cited paper also explains the grandparent exclusions (note that no subregions of the Americas are defined as reference populations – see Table 1 of cited paper), but we reproduce this information here for the reviewer's interest: "For most reference populations, the research-consented 23andMe customers reported in survey responses that their four grandparents were born in a single country. For regions with large multiethnic countries (e.g., South Asia), we also required that an individual's four grandparents either spoke a single regional language or were born in one state. Free-text responses on grandparental national, ethnic, religious, or other identities enabled us to construct reference panels for populations not defined by specific geographic regions (e.g., Ashkenazi Jews)."

Inserted text in Supplementary Methods, page 4, 2nd paragraph:

"Individuals were only included if they had > 97% European ancestry, as determined through an analysis of local ancestry (see ¹ for further details on the methodology used).."

¹ A scalable pipeline for local ancestry inference using tens of thousands of reference haplotypes. Eric Y. Durand, Chuong B. Do, Peter R. Wilton, Joanna L. Mountain, Adam Auton, G. David Poznik, J. Michael Macpherson. bioRxiv 2021.01.19.427308; doi: <https://doi.org/10.1101/2021.01.19.427308>"

Minor Concerns:

Comment 5) The authors claim in the intro that dyslexia affects 10% of the population. In their data, however, only 5% of the sample reports having dyslexia. Is this evidence of selection in the sample? If individuals with dyslexia are less likely to be part of the sample, could this lead to collider bias in the results?

Response: While we mention a midway point of up to 10% prevalence in the abstract, dyslexia prevalence has been shown to range substantially depending on definition. Even the 5% prevalence in

23andMe is consistent with the 5 to 17.5% range for dyslexia prevalence that we cite in our introduction (page 4). As the reviewer highlights, sample selection is a possibility and we hypothesised in the methods (page 3) that people with dyslexia may be less likely to participate in online survey research because of the reading demands, so there may be selection against people with dyslexia. But at the same time, 23andMe participants (and voluntary research participants more generally) tend to be of higher socio-economic status, and the presence of this selection might actually strengthen the power of our study because diagnosis of dyslexia is more common in higher SES, resulting in fewer (undiagnosed) cases among the controls. Because the outcome is a developmental trait, collider bias due to selection on adult characteristics might be reduced but it may still be present, we have no way of knowing this at present for dyslexia. We have added a sentence in the discussion on the potential for and implications of collider bias.

Inserted text in Discussion of main document, page 17, 2nd sentence:

Page 12, paragraph 2:

“However, a limitation of our study is the potential for collider bias arising from sample selection (i.e., people without dyslexia and from higher socio-economic positions) which we were unable to quantify; thus, care should be taken in future research when using polygenic scores based on many variants⁵¹”.

⁵¹ Munafò, M.R., Tilling, K., Taylor, A.E., Evans, D.M. & Davey Smith, G. Collider scope: when selection bias can substantially influence observed associations. *Int J Epidemiol* **47**, 226-235 (2018).

Comment 6) The authors report a different number of independent variants in the MR analysis (51) than they report in the GWAS (42). I think this is because they use a different locus definition in each case. Is that true? If so, why?

Response: We have removed the MR analysis so this no longer applies, but yes, the linkage disequilibrium parameters were the default ones (10,000kb window, R² threshold of p<.001) used by the MR program so differ from the more in-depth procedure used by 23andMe.

Comment 7) Related to point 6, I wasn't able to find an explanation of the locus definition in the main text, methods section, or supplement. I did find some text in the note to Figure 1 ("Variants located within a distance of < 250 kb of each other are considered as one locus."), but it seems this information should also go in the text somewhere. If the text in the figure note is complete, this seems like an odd locus definition. In other GWASs that I'm aware of, the definition generally uses a wider window and often includes an r-squared criterion. How did the authors select this criterion?

Response: We apologise for not making the locus definition more complete and salient in the manuscript. We now more fully describe this in the Methods. We do not use an r-squared criterion to define a locus because variant pruning based on LD uses an arbitrary threshold. Our approach, which is used in all 23andMe GWAS papers, permits a better description of genomic regions with many close or secondary association signals.

Inserted text in Methods of main document, page 14:

"We define associated regions by first identifying all variants with $p < 5 \times 10^{-8}$, then grouping these variants into regions separated by gaps of at least 250 kb. Index variants are the variants with smallest p -value within each associated region. If there are no variants with $p < 5 \times 10^{-8}$, we use a threshold of $p < 10^{-5}$."

Comment 8) In lines 118-119, the authors report several p -values. Are these p -values for a test against the null of $r_g=0$? Wouldn't it make more sense to report p -values for a test of $r_g=1$? Or better yet, perhaps it would be best to just report the 95% confidence intervals for these estimates so readers can get a sense of how precise the estimates are more generally. (Likewise, several genetic correlations are reported in lines 165-176, and it would be helpful if confidence intervals were reported there too.)

Response: Yes, the p -values we report are for a test against the null hypothesis of no correlation. Given that the LDSC estimates are not bounded and can thus exceed 1 due to sampling variation, we think it is problematic to use $r_g=1$ as the difference test. But thank you for the good suggestion of including 95% CIs for these described genetic correlations, which we now do.

Inserted in Main document, page 6:

"Genetic correlation estimated by LDSC was 0.91 (95% Confidence Intervals: 0.86 - 0.96; $p = 8.26 \times 10^{-253}$) in males and females, and 0.97 (95% CIs: 0.91 - 1.02; $p = 2.32 \times 10^{-268}$) between younger and older adults (Supplementary Table 16)."

Inserted in Main document, page 8:

"Genetic correlations (r_g) with quantitative reading and spelling measures ranged from -.70 to -.75 (lowest 95% CI of -.60, highest 95% CI of -.86), with phoneme awareness -.62 (95% CIs: -.50, -.74), and nonword repetition -.45 (95% CIs: -.26, -.64). Childhood/adolescent performance IQ r_g was lower (-.19; 95% CIs: -.08, -.30) than adult verbal-numerical reasoning¹⁹ (-.50; 95% CIs: -.45, -.55) but similar to childhood IQ and educational attainment²⁰ (-.32 (95% CIs: -.21, -.43) and -.22 (95% CIs: -.15, -.29), respectively). Positive r_g included: jobs involving heavy manual work²⁰ (.40; (95% CIs: .34, .45)), work-related/vocational qualifications (.50; 95% CIs: .41, .59), ADHD²¹ (.53; 95% CIs: .29, .77), equal use of right and left hands²⁰ (.38; 95% CIs: .19, .57), and pain measures²⁰ (average = .31, 95% CIs: .21, .41). Of the 11 ENIGMA measures tested, only intracranial volume was significantly correlated with dyslexia ($r_g = -.14$; 95% CIs: -.06, -.22)."

Reviewer #2:

Remarks to the Author: This is a GWAS of dyslexia with ~50K cases and ~1M controls (all from 23andMe). Forty-two genome-wide significant hits were found. The study seems to be well done and

17will be of great interest to researchers interested in mental abilities and reading. The most intriguing finding is a moderate genetic correlation between dyslexia and ambidexterity.

Response: Thank you for your compliments. We agree that the ambidexterity finding is interesting and could re-focus current thinking on laterality and influence new hypotheses for the field. We now address your comments in turn.

Comment p. 6, lines 92-93: "We identified 42 independent genome-wide significant associated loci". As far as I can tell, the paper never gives its definition of an independent genome-wide significant locus. There are many reasonable ones, and any one of these will do. But it has to be stated somewhere what was used.

Response: Reviewer 1 also made this comment, and we apologise for not highlighting this detail in the original submission. We add the following text to the Method section.

Inserted text in Methods of main document, page 14:

"We define associated regions by first identifying all variants with $p < 5 \times 10^{-8}$, then grouping these variants into regions separated by gaps of at least 250 kb. Index variants are the variants with smallest p-value within each associated region. If there are no variants with $p < 5 \times 10^{-8}$, we use a threshold of $p < 10^{-5}$."

Comment p. 6, lines 100-101: "pleiotropic with general cognitive ability based on HEIDI-outlier analysis". HEIDI is a Mendelian-randomization tool. This is the first time that I have seen it used specifically to identify pleiotropic SNPs. But I strongly recommend dropping this. Its performance depends on the statistical power of the two GWAS, which the authors seem not to consider. It requires the specification of a significance threshold (the default is, last I checked, $p < .01$), and the authors do not address whether this is optimal for their purpose. Also, the definition of pleiotropy effectively adopted by the use of HEIDI is a rather strange one; it means that the SNP is more associated with cognitive ability than expected on the basis of a model where cognitive ability affects dyslexia (according to Methods line 114). Is it absolutely necessary to exclude such SNPs from the downstream selection of likely causal genes?

Response: We thank the reviewer for highlighting concerns with the use of HEIDI. We had thought this tool could be useful to exclude variants showing horizontal pleiotropy so that our causal genes were more proximal to reading processes. We have gone back to look at the HEIDI method documentation and it seems the authors of that have developed a new detection method which indeed suggests problems with the method we used. And then of course, there is the reviewer's concern for bias due to unequal statistical power of the two GWAS: the standard error of the LDscore intercept was larger for cognitive performance (i.e., smaller N of 257,841) suggesting that this may be a problem. Therefore, we agree that the HEIDI analysis should be dropped. Because we had originally excluded SNPs from Table 1 that were pleiotropic with cognitive ability (based on HEIDI analysis), we have now re-instated a couple of SNPs as novel, so we report 27 rather than 25 novel associations in Table 1.

Comment p. 8, lines 138-140: "Gene property analysis for general tissues confirmed the importance of brain and specific regions (Supplementary Tables 13 and 14)". What does this mean? Does this correspond to Methods, lines 130-137?

Response: We can see why this was confusing since we hadn't referred to the brain specific tissue analysis in the method, nor was it clear from the main paper that this was done. We have clarified the issue in both sections.

Inserted text in Main document, page 7:

"Gene property analysis for general tissues and 13 brain tissues confirmed the importance of brain and specific brain regions (Supplementary Tables 13 and 14)."

Inserted text in Methods of Main document, page 16:

"Tissue specificity for differentially expressed genes (DEG) was tested in FUMA using the pre-calculated data sets for tissue-specific and brain region-specific DEGs."

Comment p. 8, lines 140-141: "Levels of brain expression for the 136 genes proximal to the top single variants are shown in Supplementary Table 15." This is confusing for several reasons. First, the previous paragraph spoke of 42 loci and the most probable causal genes for each one. This suggests 42 genes, although a greater number (i.e., 136) is logically consistent with what is written. Second, the word "proximal" suggests that the 136 genes are selected on basis of mere proximity to the likely causal SNPs and not by the "Overall V2G score" mentioned in the previous paragraph. Is this in fact the case? Third, regardless of how the previous two points are resolved, it seems clear that the authors have shifted from the 173 MAGMA-prioritized genes mentioned in line 135 to some other set of genes. (Or are the 136 a subset?) Readers who do not care to go back and forth between the main text and the supplementary material might be either lost or made suspicious by what seems to be arbitrary changes in the analysis. If indeed different sets of genes are being employed in different analyses, I suggest that the authors instead select one DEG method for picking out genes as the "main" method and other methods as "robustness checks", always run the main genes through the analyses (e.g., expression), and give the results of the "robustness" genes as asides or in the supplementary material. This might be a bit artificial post hoc, but it will at least help readers follow along.

Response: We agree that this is confusing, and on carefully checking these results again, we discovered some inaccuracies in our reporting. In our supplementary methods where we stated that genes proximal to the top SNP hits were functionally annotated, this was not correct; rather, it was genes from the gene-based tests. Similarly, in Supplementary Table 11, the gene function is for gene-based tests rather than for those proximal to our top SNP hits (note that we now include rows for novel proteins here, so that the table exactly matches the number (173) of significant genes from the gene-based association tests). Hence, with the exception of Supplementary Table 15, all gene-based downstream analysis is on genome-wide significant genes from gene-based tests. For Supplementary Table 15, the brain tissue

specific annotation included genes that were significant in gene-based tests (N=173 but only 125 mapped in FUMA) as well as those that were proximal (by distance) to our top SNP associations (N=55 with the two closest known genes selected for intergenic SNPs but only 45 mapped in FUMA) with overlap between these sets resulting in a combined N of 136 genes. But the reviewer's comment has made us think more about this issue and the arbitrariness of using distance to gene rather than, for instance, overall V2G score. To keep it simpler for the reader, we have now removed 11 'proximal' genes from this Supplementary Table 15 that did not overlap with the gene-based test genes (these were indicated by a single * in the original version of this table).

Inserted text in Supplementary Methods, page 8:

"And the following annotations of genes which were significant in genome-wide gene-based tests (Supplementary Table 12):"

Updated table heading in Supplementary Table 11 (and added function for several that were missing due to archived aliases):

"Descriptions of the Functions of Genes (where known) that are Significantly Associated with Dyslexia in Genome-wide Gene-based Tests"

Re-arranged sentence placement and inserted text in Main document, page 7:

"For the 173 significant genes from gene-based tests, there was enrichment of differentially expressed genes in brain subcortical and cortical areas (Supplementary Figure 6). Levels of brain expression for 125 of these genes could be mapped in FUMA and are shown in Supplementary Table 15. Twenty genes showed high general brain expression levels (of these, PPP1R1B, NPM1, and WASF3 were located near significant SNP associations)."

Comment p. 9, line 175: "spectral decomposition of matrices". What does this mean?

Response: We apologise for the lack of detail here. We now mention the PhenoSpD software we used, briefly describe the method, and provide the relevant citation.

Inserted text in Main document, page 8:

"Targeted investigation of 80 UK Biobank structural neuroimaging measures including surface-based morphometry and diffusion-weighted imaging for brain circuitry linked to language were non-significant at a Bonferroni-corrected significance level for number of independent traits. Phenotype independence was estimated by spectral decomposition of the phenotypic correlation matrix implied by the bivariate LDscore regression intercept from GWAS summary statistics of these traits, using the PhenoSpD toolkit²²."

²² Zheng, Jie et al. "PhenoSpD: an integrated toolkit for phenotypic correlation estimation and multiple testing correction using GWAS summary statistics." *GigaScience* vol. 7,8 giy090. 1 Aug. 2018,

20doi:10.1093/gigascience/giy090

Comment pp. 9-10, Mendelian randomisation: Although MR has become standard in GWAS papers, this is a huge mistake. MR is simply not a reliable method for causal inference. See the clear statement in Bowden et al. (2015), the paper introducing MR-Egger: "The InSIDE assumption would be violated if the pleiotropic effects act via a confounder of the exposure-outcome association." But this is precisely the case if there are heritable confounders or the causal direction is backward. If the authors want to do an MR "type" method, why not the ingenious LCV method of O'Connor and Price (2018)? I suspect that the LCV method is not more popular because it often does not deliver the results desired. (Incidentally the simulations reported in this paper show that Egger, median, and mode methods all do poorly.) The authors might be interested in the CAUSE method of Morrison et al. (2020), which if I understand correctly can discern a true causal relationship in certain situations where LCV cannot. If the authors do not want to bother with new analyses, then I suggest entirely omitting attempts to put a causal order to two genetically correlated variables. In the past, however, editors and reviewers have not followed my suggestion about this matter. If that happens again, then I ask that the caveat in the Discussion ("the result is tentative given the potential for unmeasured confounding with the genetic instrument," p. 14, lines 267-268) be stated earlier in the Results and later in the Methods, loud and clear.

Response: Many thanks for pointing us to the CAUSE method and for encouraging us to use LCV. On the editors' request (and see reviewer 1's comment 1), we have removed the MR analysis from the paper. We do think the causal questions are really interesting and could potentially have practical implications, but we agree that we need a more cautious and thorough analysis of the data. With respect to ADHD, it would be prudent to wait for a similarly powered GWAS to enable an unbiased evaluation of causal directions and using the recommended methods to minimise confounding.

Comment p. 12: "Consistent with studies of other neurodevelopmental disorders, partitioning of SNP-based heritability revealed enrichment in conserved regions". As far as I know, such enrichment has been found for *all* traits in GWAS, not just neurodevelopmental disorders. To imply that dyslexia falls into a special group with other neurodevelopmental disorders is quite misleading.

Response: Yes, it's true that all complex traits have been found to be enriched in conserved regions, but we were attempting to point out that these neurodevelopmental studies found enrichment in both conserved genes and in specific histone marks. We appreciate that this wasn't clear and have re-written the sentence, adding a citation for the finding that enrichment in conserved regions is typical of human complex traits generally.

Re-written text in Main document, page 10:

"Like other human complex traits, partitioning of SNP-based heritability revealed enrichment in conserved regions³⁸, but we further observed enrichment in the histone mark H3K4me1 (as was found

for ASD³⁹), and at H3K4me1 and H3K4me3 clusters in the CNS (marking enhancers and promoters respectively).”

³⁸ Finucane HK, Bulik-Sullivan B, Gusev A, Trynka G, Reshef Y, Loh PR, Anttila V, Xu H, Zang C, Farh K, Ripke S. Partitioning heritability by functional annotation using genome-wide association summary statistics. *Nature genetics*. 2015 Nov;47(11):1228-35

Comment: Methods, lines 103-104: This is more of a suggestion than a prescription. I hesitate to tell authors to do some other analysis than the one they did when the alternative analysis is better, but it is unclear how much better or whether it will lead to substantially different results. But I think that the Maller method of picking out likely causal SNPs is outdated because it assumes only one causal SNP in the locus and ignores functional information. A better method is that of Weissbrod et al. (2020), which seems very feasible for these authors because all of their data comes from a single cohort (23andMe) whose LD might be estimated by them.

Response: Many thanks for highlighting the method of Weissbrod et al (2020). We are familiar with various alternative fine mapping methodologies to Maller’s, but given that there is no strong evidence that these new approaches perform better when only one causal variant is involved, we still see value in Maller’s method. In theory, it would be valuable to identify all causal haplotypes (and their variant composition) in an associated locus. In practice, although many associated loci show secondary signals, the vast majority of them are characterized by a unique association (only one causal haplotype or multiple causal haplotypes but with one that predominates). In addition, the new fine mapping approaches have their own limitations (some of which are highlighted in Weissbrod et al (2020)): they are sensitive to subtle population stratification and can generate false positive results in regions with very complex LD structure or with low statistical power. The functional priors add many assumptions to the models which might not be relevant and interpretable in the context of a specific trait or disease (priors are generally not tissue- or ancestry-specific, for example). Because our manuscript focusses on the genetic architecture of dyslexia and not on the fine mapping of the known or newly discovered associated loci, it seems more appropriate to present inclusive credible sets produced by a methodology with a limited set of assumptions. Interested readers could then refine the credible sets by applying functional information.

Comment: Methods, Partitioned heritability: Only Finucane et al. (2015) is cited for stratified LD Score regression. But it seems that the authors used baseline annotations introduced after the publication of that paper (e.g., Gazal et al., 2017). To the extent that additional annotations coincide with published papers, I think those papers should be cited.

Response: We thank the reviewer again for their careful reading of our paper, they are right in that the Gazal et al., 2017 reference should have been cited and not Finucane et al., 2015. We have corrected this.

Changed text in Methods of Main document, page 8:

“Enrichment of heritability was estimated in adult brain Human Gained Enhancers (HGEs) [24], foetal brain HGEs [25], ancient selective sweep regions [26], Neanderthal-introgressed SNPs [27] and Neanderthal-depleted regions [28] (see Supplementary Materials for a description of each annotation); and controlled for using the baselineLD v2 model from Gazal et al.,⁷⁵.”

Comment: Methods, line 189: I am interested in the Eising preprint but cannot find it. I think that preprints should not be treated differently from published papers in the references; please provide sufficient information for a reader to find the preprint.

Response: We apologise for not updating the Eising et al., BioRxiv reference which unfortunately wasn't uploaded in time for inclusion. We should have replaced this placeholder with a published abstract. The pre-print is now available on BioRxiv, as follows:

Eising, E., et al. (2021). "Genome-wide association analyses of individual differences in quantitatively assessed reading- and language-related skills in up to 34,000 people." [bioRxiv: 2021.2011.2004.466897](https://doi.org/10.1101/2021.2011.2004.466897).

Comment: Methods, line 229: A paper by Hill et al. (2016) is given as authority for a genetic correlation between childhood and adult IQ of 0.71. But according to the paper, this has a standard error of 0.10. A later paper by Sniekers et al. (2017) gives the genetic correlation as 0.89 with a standard error of 0.08. True, the SE is not that much smaller, but it is probably small enough to exclude 0.71 from the 95% CI. Why not cite both papers and give the range 0.7-0.9.

Response: Thank you for alerting us to this finding that the childhood-adult IQ genetic correlation is even larger than .71. On request of the editors, we have now removed the section on MR and so the manuscript no longer refers to this correlation.

Comment: Methods, line 246: "PRSice". Again, a suggestion and not a prescription. Why not a method that uses LD to convert GWAS summary statistics into partial regression coefficients (e.g., Prive et al., 2020). The resulting prediction R² will be at least a little better.

Response: Again, thank you for recommending potentially superior analysis software. Polygenic prediction is not the central aim of this study, but we do intend to focus future research on diagnostic utility of the dyslexia PGS and we will use LDpred2 for this.

Reviewer #3:

General Comments:

The authors of this interesting study are to be commended on their undertaking of studying the genetics of dyslexia in their uniquely powered questionnaire-based dataset

Response: We thank the reviewer for this positive comment.

The low prevalence of dyslexia in this sample speaks to the probably high-functioning of diagnosed dyslexics who participate in 23andMe's studies. This case/control definition does not rule out, but may lower prevalence of undiagnosed dyslexia/reading difficulties from controls.....

Response: Reviewer 1 (comment 5) asked a question about the low prevalence and part of our reply might be relevant here. We noted that 23andMe participants (and voluntary research participants more generally) tend to be of higher socio-economic status, and that this might actually strengthen the power of our study because diagnosis of dyslexia is more common in higher SES, so there might indeed be fewer (undiagnosed) cases among the controls.

The other 25 loci are considered by the authors novel associations with dyslexia, as they have no previous GWAS associations with cognitive traits. This classification is debatable. However, the number of significant dyslexia associations, pleiotropic with cognitive traits or not, is a significant contribution to the genetics of dyslexia, of which little is previously known.

Response: Thanks again for recognising the significance of our work. Just to clarify, our classification of novel SNP associations was really to highlight that dyslexia is not only dependent on so-called "generalist genes" i.e. those for which variants impact on general aspects of cognition. The associations with SNPs previously linked to cognitive ability are also novel for dyslexia and will be equally as important to follow-up. On the other hand, it is likely that efforts are already underway to better understand the effects of such SNPs since they are implicated in cognitive traits in the prior literature.

Specific Comments:

Comment 1: Overall, the GWAS methodology is well described and appropriate, although the conventional threshold $p < 5 \times 10^{-8}$ is somewhat relaxed for novel discovery. How many variants were tested?

Response: We are glad that you found our methodology well described and appropriate. We tested 57.5M variants and 23.3M variants passed the different quality controls. Finally, we applied a MAF>1% filter for a final set of 7,995,923 common variants. The significance threshold for genome-wide significance is standard for GWAS genome discovery, and has been deemed suitable for common variants in imputation panels like 1000G (Li et al 2012).

Li, Miao-Xin et al. "Evaluating the effective numbers of independent tests and significant p-value thresholds in commercial genotyping arrays and public imputation reference datasets." *Human genetics* vol. 131,5 (2012): 747-56. doi:10.1007/s00439-011-1118-2

Comment 2: The authors validate their results in samples in (3) word-reading and spelling traits (33,959 and 18,514 respectively) from GWAS meta-analysis of children and adolescents from Europe, USA and Australia (GenLang consortium) which is referenced under Eising et al. bioRxiv (line 106 p. 6). This reference for the largest replication cohort credited with the most replications in Table 1, was, however, not found in the manuscript references nor in bioRxiv.

24Response: We apologise for not updating the Eising et al., BioRxiv reference which unfortunately wasn't uploaded in time for inclusion. We should have replaced this placeholder with a published abstract. The pre-print is now available on BioRxiv, as follows:

Eising, E., et al. (2021). "Genome-wide association analyses of individual differences in quantitatively assessed reading- and language-related skills in up to 34,000 people." [bioRxiv: 2021.2011.2004.466897](https://doi.org/10.1101/2021.2011.2004.466897).

Comment 3: Table 1 (in title: novel associations rather than novel variants) presents results for the 25 defined dyslexia loci from the discovery cohort. Effect allele is listed and frequency (presumably of effect allele?), but inconsistently presented is the risk allele or minor allele which is confusing. It would be better to select as effect allele, either always the minor allele, or always the risk allele. Validation cohort is presented in the last column with p-value only, but missing from the table are validation phenotype, size of validation sample and validation effects. Furthermore, the authors use $p < 0.05$ for replication significance. However, if 42 variants are tested for replication in 4 phenotypes (Supplementary Tables 3-6), the threshold for combined replication analysis requires further adjustment.

Response: Thank you for the better suggestion of the Table name, which we have now changed. Note that we now report 27 novel associations because removal of our HEIDI pleiotropy analysis (at the request of reviewer 2) means that there are additional variants with no confirmed overlap with cognitive ability.

We agree that having an inconsistent presentation of risk in Table 1 is confusing and have now standardised this so that the risk allele is always presented. In doing so, we noticed some errors in determining the consistency of effect direction in our validation samples (we had inadvertently taken the effect allele as the risk allele, which was not always the case). After addressing these errors, it becomes clear that almost all validation sample effects are in the consistent direction, providing support that these represent true associations.

With regard to the validation statistics presented, we kept this information to a minimum in the main manuscript so that the table was easier to digest, but note that N, effect size, SE and other details are shown for all significant SNPs in the Supplementary Tables 3-6. The supplementary tables also highlight the Bonferroni-corrected significance level where we correct for number of SNPs within each cohort. We also mention in the text (page 5) whether or not replications are attained at the corrected level. Thus, we provide all the relevant information in a transparent way. Given that we have multiple validation samples, we feel that converging evidence from these samples, even at a nominal (.05) significance level, is encouraging.

Comment 4: In Supplementary Tables 3 (GenLang reading ability – assume higher score is better?) and 4 (GenLang spelling ability – assume higher score better?), some SNPs are marked with *Consistent effect direction. Consistent with what?

Response: Thank you for pointing out our lack of detail about the intended meaning of “consistent” in this context. For the Chinese and GenLang cohorts, quantitative traits were measured in the direction of higher scores relating to better reading skills. Therefore, a consistent direction of allelic effect for these reading measures would be the reverse effect of dyslexia risk (i.e., the allele coding for higher dyslexia risk ($OR > 1$) codes for lower reading scores (negative beta), and vice versa). The NeuroDys validation is for dyslexia, so the risk allele would need to be the same to be considered consistent. We have clarified this in the table footnotes.

Revised footnote for Supplementary Tables 3, 4, and 6:

“ Consistent effect direction (i.e., allele conferring increased risk of dyslexia is associated with a lower reading score, and vice-versa)”*

Revised footnote for Supplementary Table 5:

“ Consistent dyslexia effect allele”*

Comment 5: The sex-specific GWAS is very interesting but results are only presented in Manhattan plots and genetic correlations. Authors state that sex-specific results show high consistency with the main GWAS. Yet the sex-specific Manhattan plots (Supplementary Figures 2 and 3) show that top associations for the sexes are quite different with males driving some signals (e.g. ARFGF2) and females others (e.g. PP2R3A). These sex differences merit more attention in the manuscript.

Response: It is true that some different associations appear between the female and male GWAS, but this is likely a function of increased power in the female sample where 17 SNPs versus 6 SNPs in the male sample were genome-wide significant. When sexes are combined (the main GWAS), all the significant sex-specific loci, bar one from the female GWAS, are detected in the combined analysis. Thus, there is nothing particularly novel to present from the sex-specific analyses. But this information is important for readers to know, so we thank the reviewer for raising the issue. We now include details about the overlap of top SNPs from the sex-specific and main GWAS, highlighting the single SNP/locus that was significant in the female GWAS but not in the main GWAS. It is an intergenic SNP with no previous GWAS associations reported.

Inserted text in Main document, page 6:

“Of the 17 genome-wide significant variants in the female GWAS, all but four (rs61190714, rs4387605, rs12031924, and rs57892111) were significant in the main GWAS. And of these four, three were in LD with a SNP that approached significance ($p < 3.3 \times 10^{-7}$ or smaller) in the main analysis. Intergenic SNP rs57892111 (located between TFAP2B and PKHD1 on chromosome 6p) was not among the main analysis significant or suggestive SNPs of the main analysis, and so may represent a female-specific variant. It has not been associated with any others traits from GWAS. Of the six genome-wide significant variants in the male GWAS, all were significant in the main GWAS.”

Comment 6: The genetic correlations with dyslexia depicted in Figure 2 and Supplementary Table 22 show results for 98 traits but unclear is how the authors selected these traits from the many more traits available in LD Hub? Traits of reading, spelling, intelligence, and handedness are justified by the authors, as are the phenotypes representing known psychiatric comorbidities. But why did the authors decide to select a set of pain phenotypes? As they state in the discussion, the correlation of dyslexia with pain was unexpected? The reference provided for pain correlation is [23] Bycroft, C., et al., The UK Biobank resource with deep phenotyping and genomic data. *Nature*, 372 2018. 562(7726): p. 203-209, indicating that many more phenotypes were available for testing.

Response: In selecting traits for the genetic correlations, we attempted to get a broad coverage of different types of traits covering health, cognitive abilities, disorders, SES, and lifestyle. We selected variables that have been genetically associated in GWAS of variables related to dyslexia, like ADHD, cognitive ability, and educational attainment. Rather than an exploratory approach of testing hundreds of variables in UKB and LDHub (which may indeed uncover novel findings) we felt that this more targeted approach would create more impact since it is easier to compare correlations from different categories when you don't have to scroll through hundreds of variables, most of which are not relevant. With regard to pain, we consider it as a health trait and indeed recent reports (Johnston et al, 2019; Meng et al, 2020) have shown genetic correlations between pain and various psychiatric, autoimmune and anthropometric traits which makes it of even greater interest. We are not experts in pain so took an inclusive approach to selection of variables in this domain, but indeed, we could have chosen a couple to be representative and classified them under the neurological trait category. Given that we have now tested these, for transparency, they should all be reported.

Johnston KJA, Adams MJ, Nicholl BI, Ward J, Strawbridge RJ, et al. (2019). Genome-wide association study of multisite chronic pain in UK Biobank. *PLOS Genetics* 15(6): e1008164.

<https://doi.org/10.1371/journal.pgen.1008164>

Meng, W., Adams, M.J., Reel, P. *et al.* Genetic correlations between pain phenotypes and depression and neuroticism. *Eur J Hum Genet* 28, 358–366 (2020). <https://doi.org/10.1038/s41431-019-0530-2>

Comment 7: Finally, under data availability the authors state that the "full summary statistics for each dyslexia GWAS presented in this paper will be made available through 23andMe to qualified researchers under an agreement with 23andMe that protects the privacy of the 23andMe participants". All research groups are equally mindful of privacy protection of their study participants, yet make summary statistics more readily available than 23andMe. Responsibly making data available to other research groups is the policy of Nature Genetics and other comparable journals and should apply to 23andMe as well.

Response: 23andMe have a simple process for accessing their GWAS summary statistics and here we follow the standard legal language used in 23andMe publications, including for Nature Genetics (see

<https://www.nature.com/articles/s41586-020-03065-y#data-availability>). However, we will make available the top 10,000 SNP associations from our GWAS, as allowed by 23andMe.

Inserted text in Data Availability Statement (page 24):

"The top 10,000 associated SNPs from the GWAS can be downloaded from <https://datashare.ed.ac.uk/>."

Decision Letter, first revision:

20th December 2021

Dear Michelle,

Your revised Article "Discovery of 42 genome-wide significant loci associated with dyslexia" has been seen by the original referees. You will see from their comments below that, while Reviewers #1 and #3 are satisfied with the revisions, Reviewer #2 has a few remaining concerns. We remain interested in publishing your study in Nature Genetics, but we would like to consider your response to these remaining concerns in the form of a further revision before we make a final decision regarding publication.

As before, to guide the scope of the revisions, the editors discuss the referee reports in detail within the team, including with the chief editor, with a view to identifying key priorities that should be addressed in revision, and sometimes overruling referee requests that are deemed beyond the scope of the current study. In this case, we ask that you carefully revise the presentation throughout for clarity and that you address Reviewer #2's concerns regarding the MAGMA analyses by performing further validation of the annotations used or by removing these analyses. We again hope you will find this prioritized set of referee points to be useful when revising your study. Please do not hesitate to get in touch if you would like to discuss these issues further.

We therefore invite you to revise your manuscript again taking into account all reviewer and editor comments. Please highlight all changes in the manuscript text file. At this stage, we will need you to upload a copy of the manuscript in MS Word .docx or similar editable format.

We are committed to providing a fair and constructive peer-review process. Do not hesitate to contact us if there are specific requests from the reviewers that you believe are technically impossible or

28unlikely to yield a meaningful outcome.

*2) If you have not done so already please begin to revise your manuscript so that it conforms to our Article format instructions, available [here](http://www.nature.com/ng/authors/article_types/index.html). Refer also to any guidelines provided in this letter.

[REDACTED]

We again hope to receive your revised manuscript within 4-8 weeks. If you cannot send it within this time, please let us know.

Sincerely,
Kyle

Kyle Vogan, PhD
Senior Editor
Nature Genetics
<https://orcid.org/0000-0001-9565-9665>

Referee expertise:

Referee #1: Genetics, complex traits, statistical methods

Referee #2: Genetics, behavioral traits, statistical methods

Referee #3: Genetics, complex traits, dyslexia

Reviewers' Comments:

Reviewer #1:
Remarks to the Author:

I have reviewed the authors' response letter and the revised manuscript, and I am satisfied with their revisions. This is a very nice paper that represents an important contribution to the literature.

Reviewer #2:

Remarks to the Author:

In addressing my concerns, the authors unfortunately raise some new ones. I hope that these can be resolved satisfactorily. A general comment is that I am bothered by the lack of professionalism in the writing of the manuscript. There are numerous malformed sentences, such as "It has not been associated with any others traits from GWAS" and "Enrichment was seen in enhancer and promoter regions with the chromatin marks H3M4me1 and H3K4me3 chromatin marks in multiple central nervous system (CNS) tissues." I worry that sentences like these reveal a lackadaisical approach to this paper, an attitude that since this is a GWAS it can be put together by the numbers. Are there similarly serious substantive problems that I am missing? In any case the authors need to take responsibility for this paper, treating every element as reflecting well or poorly on them as individuals.

Methods, p. 14, lines 309-312: "We define associated regions by first identifying all variants with $p < 5 \times 10^{-8}$, then grouping these variants into regions separated by gaps of at least 250 kb. Index variants are the variants with smallest p-value within each associated region. If there are no variants with $p < 5 \times 10^{-8}$, we use a threshold of $p < 10^{-5}$."

I don't think I've ever seen this definition used before. I think it is fine since it seems conservative in the sense of leading to fewer significant loci than other possible approaches. But what does the last sentence mean?

p. 5, lines 86-87: Maybe related to the previous point. What use is made of the "64 loci with suggestive significance ($p < 1E-6$)"?

p. 7, lines 136-138: "For the 173 significant genes from gene-based tests, there was enrichment of differentially expressed genes in brain subcortical and cortical areas (Supplementary Figure 6)." Now that some things are more clearly explained, I have still more questions. It is not mentioned in the main text that the most significant tissue in this analysis is the heart. This makes me wonder whether this analysis is sound. Have other GWAS used these gene-level annotations with MAGMA? I would normally think that you could use anything with MAGMA, but these strange results make me wonder these

31annotations should be further validated, perhaps by simulations where the causal SNPs are chosen at random, in silico GWAS are conducted, and the results are subjected to this gene-set analysis with MAGMA. A tendency for heart to be significant in this case would show that there is something wrong.

I am not making a specific recommendation here. I am just pointing out what seems to me an anomaly. It also seems that this analysis is not necessary to establish which tissue or organ (likely the brain) is the site of action; Supplementary Figures 10 and 11 suggest this pretty well, I believe.

A comment about the Supplementary Figures. Each figure and table should have a caption that is close to self-contained, so that consulting the text or other part of the paper is not absolutely necessary to understand the figure/table. This principle is not followed by the authors. I suggest that they make the attempt.

Reviewer #3:

Remarks to the Author:

I find the authors' responses to the issues raised by reviewer's satisfactory and the edited manuscript improved.

Author Rebuttal, first revision

We thank the editors for their guidance and reviewer 2 for their further comments on our manuscript which we have found useful. We hope that our changes now allay any remaining concerns. Most notably, we have removed the queried MAGMA analysis in line with the editors' recommendation and have carefully read the manuscript to check our clarity in writing and expression. We give more detailed information about these changes below in our response to the reviewers' comments.

Reviewers' Comments:

Reviewer #1:

Remarks to the Author:

I have reviewed the authors' response letter and the revised manuscript, and I am satisfied with their revisions. This is a very nice paper that represents an important contribution to the literature.

Response: We thank the reviewer for their positive comment on our work.

Reviewer #2:

Remarks to the Author:

In addressing my concerns, the authors unfortunately raise some new ones. I hope that these can be resolved satisfactorily. A general comment is that I am bothered by the lack of professionalism in the writing of the manuscript. There are numerous malformed sentences, such as "It has not been associated with any others traits from GWAS" and "Enrichment was seen in enhancer and promoter regions with the chromatin marks H3M4me1 and H3K4me3 chromatin marks in multiple central nervous system (CNS) tissues." I worry that sentences like these reveal a lackadaisical approach to this paper, an attitude that since this is a GWAS it can be put together by the numbers. Are there similarly serious substantive problems that I am missing? In any case the authors need to take responsibility for this paper, treating every element as reflecting well or poorly on them as individuals.

Response: We thank the reviewer for pointing out the presence of some malformed sentences in the text of our prior revision. We aimed for concise writing throughout, to convey our findings within the strict word limits of the journal, but it seems that in doing this we unintentionally lost clarity in some parts of the manuscript, and a handful of grammatical errors slipped through. We can reassure the reviewer that this does not reflect a "lackadaisical approach". Indeed, far from "putting it together by the numbers", this paper is the result of considerable hard work and a great deal of investment in time and effort by the co-authors of the work over multiple years. The other reviewers commented on our paper being "well written" and noted that the "methodology is well described". Nevertheless, we have re-read our manuscript very carefully for clarity and made relevant changes to wording/sentence structure (as marked in the revised manuscript), including to the two sentences flagged which we highlight below:

"It has not been associated with any others traits from GWAS."

changed to:

"There is no evidence from existing GWAS that this SNP is associated with any other human trait."

“Enrichment was seen in enhancer and promoter regions with the chromatin marks H3M4me1 and H3K4me3 chromatin marks in multiple central nervous system (CNS) tissues.”

changed to:

“Enrichment was seen in enhancer and promoter regions, identified by the presence of H3M4me1 and H3K4me3 chromatin marks, respectively, in multiple central nervous system (CNS) tissues”

Comment: Methods, p. 14, lines 309-312: "We define associated regions by first identifying all variants with $p < 5 \times 10^{-8}$, then grouping these variants into regions separated by gaps of at least 250 kb. Index variants are the variants with smallest p-value within each associated region. If there are no variants with $p < 5 \times 10^{-8}$, we use a threshold of $p < 10^{-5}$."

I don't think I've ever seen this definition used before. I think it is fine since it seems conservative in the sense of leading to fewer significant loci than other possible approaches. But what does the last sentence mean?

Response: The last sentence was trying to describe the approach for regions with suggestive association(s) but it was definitely unclear. We replaced the last sentence with:

"We use the same approach for regions with suggestive associations, but by first identifying all variants with $p < 10^{-5}$."

Comment: p. 5, lines 86-87: Maybe related to the previous point. What use is made of the "64 loci with suggestive significance ($p < 1E-6$)"?

Response: None of the downstream analyses directly focused on the suggestive associations, although gene-based analyses, for example, are strongly driven by the single variant suggestive associations. It is common practice to publish suggestive associations for two main reasons: a) it often makes it possible to address the question of replication of previously published associations (in our case, none of the previously-reported variants were replicated); and b) it offers the reader easy access to association statistics for variants of interest (identified by independent studies on other relevant phenotypes or PheWAS-related questions).

Comment p. 7, lines 136-138: "For the 173 significant genes from gene-based tests, there was enrichment of differentially expressed genes in brain subcortical and cortical areas (Supplementary Figure 6)." Now that some things are more clearly explained, I have still more questions. It is not

34mentioned in the main text that the most significant tissue in this analysis is the heart. This makes me wonder whether this analysis is sound. Have other GWAS used these gene-level annotations with MAGMA? I would normally think that you could use anything with MAGMA, but these strange results make me wonder these annotations should be further validated, perhaps by simulations where the causal SNPs are chosen at random, in silico GWAS are conducted, and the results are subjected to this gene-set analysis with MAGMA. A tendency for heart to be significant in this case would show that there is something wrong.

I am not making a specific recommendation here. I am just pointing out what seems to me an anomaly. It also seems that this analysis is not necessary to establish which tissue or organ (likely the brain) is the site of action; Supplementary Figures 10 and 11 suggest this pretty well, I believe.

Response: On further investigation, we found that 70% of the top genes expressed in the brain are also expressed in the heart, and 51% of the top genes expressed in the heart are expressed in the brain. Fifteen of our significant SNPs are associated with blood pressure and 5 with coronary heart disease from previous GWAS. It is likely then that focussing on our top genes, many of which happen to also be related to cardiology traits, may have biased the results of the DEG analysis. A more robust gene property analysis which evaluates expression based on all our association results (Supplementary Table 13) shows the brain as the most significant tissue, while the heart is non-significant. As per Editors' guidance we have removed the DEG analysis from the paper.

Comment: A comment about the Supplementary Figures. Each figure and table should have a caption that is close to self-contained, so that consulting the text or other part of the paper is not absolutely necessary to understand the figure/table. This principle is not followed by the authors. I suggest that they make the attempt.

Response: We aimed for our captions to be as informative as possible, but agree that some are better than others. We have changed the following in the Supplementary Figures:

All supplementary figures 4.i-xliii have been changed as per following example:

“Supplementary Figure 4.ii. Regional association plot for chr3q22.3 rs13082684”

Changed to:

“Supplementary Figure 4.ii. Regional association plot which includes information on the credible variant set and known genes in the region for the significant association between dyslexia and chr3q22.3 rs13082684”

Also note the revised table captions in the Supplementary Tables file, which are highlighted in green.

Reviewer #3:

Remarks to the Author:

I find the authors' responses to the issues raised by reviewer's satisfactory and the edited manuscript improved.

Response: We are pleased that the reviewer sees improvement in our manuscript and that we have successfully addressed concerns raised through the review process.

Decision Letter, second revision:

Our ref: NG-A58332R1

30th March 2022

Dear Michelle,

Thank you for submitting your revised manuscript "Discovery of 42 genome-wide significant loci associated with dyslexia" (NG-A58332R1). In light of the changes made in responses to Reviewer #2's comments and the positive feedback from the earlier rounds of review, we will be happy in principle to publish your study in Nature Genetics as an Article pending final revisions to comply with our editorial and formatting guidelines.

We are now performing detailed checks on your paper and we will send you a checklist detailing our editorial and formatting requirements soon. Please do not upload the final materials and make any

36revisions until you receive this additional information from us.

Sincerely,
Kyle

Kyle Vogan, PhD
Senior Editor
Nature Genetics
<https://orcid.org/0000-0001-9565-9665>

Final Decision Letter:

In reply please quote: NG-A58332R2 Luciano

23rd August 2022

Dear Michelle,

I am delighted to say that your manuscript "Discovery of 42 genome-wide significant loci associated with dyslexia" has been accepted for publication in an upcoming issue of Nature Genetics.

37You will not receive your proofs until the publishing agreement has been received through our system.

Your paper will be published online after we receive your corrections and will appear in print in the next available issue. You can find out your date of online publication by contacting the Nature Press Office (press@nature.com) after sending your e-proof corrections. Now is the time to inform your Public Relations or Press Office about your paper, as they might be interested in promoting its publication. This will allow them time to prepare an accurate and satisfactory press release. Include your manuscript tracking number (NG-A58332R2) and the name of the journal, which they will need when they contact our Press Office.

Before your paper is published online, we will be distributing a press release to news organizations worldwide, which may very well include details of your work. We are happy for your institution or funding agency to prepare its own press release, but it must mention the embargo date and Nature Genetics. Our Press Office may contact you closer to the time of publication, but if you or your Press Office have any enquiries in the meantime, please contact press@nature.com.

Acceptance is conditional on the data in the manuscript not being published elsewhere, or announced in the print or electronic media, until the embargo/publication date. These restrictions are not intended to deter you from presenting your data at academic meetings and conferences, but any inquiries from the media about papers not yet scheduled for publication should be referred to us.

Please note that Nature Genetics is a Transformative Journal (TJ). Authors may publish their research with us through the traditional subscription access route or make their paper immediately open access through payment of an article-processing charge (APC). Authors will not be required to make a final decision about access to their article until it has been accepted. . Please let your coauthors and your institutions' public affairs office know that they are also welcome to order reprints by this method.

If you have not already done so, we invite you to upload the step-by-step protocols used in this

manuscript to the Protocols Exchange, part of our on-line web resource, natureprotocols.com. If you complete the upload by the time you receive your manuscript proofs, we can insert links in your article that lead directly to the protocol details. Your protocol will be made freely available upon publication of your paper. By participating in natureprotocols.com, you are enabling researchers to more readily reproduce or adapt the methodology you use. Natureprotocols.com is fully searchable, providing your protocols and paper with increased utility and visibility. Please submit your protocol to <https://protocolexchange.researchsquare.com/>. After entering your nature.com username and password you will need to enter your manuscript number (NG-A58332R2). Further information can be found at <https://www.nature.com/nature-portfolio/editorial-policies/reporting-standards#protocols>

Sincerely,
Kyle

Kyle Vogan, PhD
Senior Editor
Nature Genetics
<https://orcid.org/0000-0001-9565-9665>